# Direct microsecond wide-field single-molecule tracking and super-resolution mapping via CCD vertical shift

**Megan A. Steves** ● & **Ke Xu** ● ✉

Wide-field single-molecule tracking is often limited by the ~10 ms camera frame time. We introduce SpeedyTrack, which directly enables microsecond wide-field single-molecule tracking/imaging on common setups. Harnessing the native sub-microsecond vertical charge shifting capability of EM-CCDs, SpeedyTrack staggers wide-field single-molecule images along the CCD chip at ~10-row spacings between consecutive timepoints, effectively projecting the time domain to the spatial domain. Wide-field tracking is achieved for molecules diffusing at up to 1000 $\mu m^2/s$ at 50 $\mu s$ temporal resolutions for >30 timepoints. Concurrent Förster resonance energy transfer measurements further elucidate molecular states. By implementing temporally patterned vertical shifting, VS-SpeedyTrack next deconvolves the spatial and temporal information to map trajectories at the super-resolution level, resolving the diffusion mode of a fluorescent protein in live cells with nanoscale resolution. Without modifications to existing optics or electronics, SpeedyTrack provides a facile solution to the microsecond tracking of single molecules and their super-resolution mapping in the wide field.

Single-particle/single-molecule tracking (SPT) is a key tool for quantifying molecular motion in cells and in vitro[1–3]. Wide-field SPT allows many molecules to be tracked simultaneously, and recent integration with photoactivation and fluorophore exchange, common strategies in single-molecule localization microscopy (SMLM)[4], further permits high-density sampling and super-resolution mapping[5,6]. More generally, the mass integration of localization, motion, spectra, and other single-molecule measurements in the wide field enables the super-resolution mapping of physicochemical parameters and molecular interactions at the nanoscale[7].

However, wide-field SPT is often limited to the slow (< 10 $\mu m^2/s$) diffusion of molecules bound to membranes, chromosomes, or the small volume of bacteria, partly owing to the ~10 ms framerate of common single-molecule cameras like electron-multiplying charge-coupled devices (EM-CCDs). For unbound diffusion in the mammalian cell and in solution, a molecule readily diffuses out of the < 1 $\mu m$ focal range of high-numerical-aperture objective lenses in the 10 ms timeframe. Although recent advances, e.g., ultrahigh-speed intensified

CMOS cameras[8], have achieved temporal solutions of < 100 $\mu s$, they are limited by specialized hardware and low light-use efficiencies, and tracking has not been demonstrated for unbound molecules. Meanwhile, by locking onto a molecule with feedback control, longitudinal tracking of unbound molecules may be achieved at ~100 $\mu s$ temporal resolutions[9–12]. However, with complicated implementations, it remains challenging to track molecules diffusing faster than ~20 $\mu m^2/s$. While single-molecule trapping detects fast diffusion[13], diffusion is no longer free, and application to cell imaging is unlikely. More generally, tracking one molecule at a time limits throughput and hinders statistics and spatial mapping. We recently introduced single-molecule displacement/diffusivity mapping (SM$d$M), which, via tandem excitation pulses across paired camera frames, overcomes the framerate limit to detect transient molecular motion in the wide field[14,15]. However, each molecule is only observed for two timepoints to determine its displacement in a single timestep. The lack of trajectories obscures diffusion evaluation, which relies on the statistics of many molecules under a simplified model.

Department of Chemistry, University of California, Berkeley, Berkeley, California, USA. ✉e-mail: xuk@berkeley.edu

Here we introduce spatially-encoded dynamics tracking (SpeedyTrack), which directly enables microsecond wide-field single-molecule tracking/imaging on standard EM-CCDs. Capitalizing on the intrinsic fast vertical shifting of EM-CCD, SpeedyTrack staggers wide-field single-molecule images along the CCD chip at ~10-row spacings between consecutive timepoints, effectively projecting the time domain to the spatial domain. We thus demonstrate the tracking of freely diffusing molecules at down to 50 μs temporal resolutions for > 30 timepoints, which uniquely permits trajectory analysis to quantify diffusion coefficients of > 1000 μm²/s. Integration with single-molecule Förster resonance energy transfer (smFRET)[16,17] further elucidates conformational dynamics and binding states for freely diffusing molecules. Moreover, we develop a temporally patterned vertical shifting scheme to deconvolve the spatial and temporal information to map fast single-molecule trajectories at the super-resolution level, thus elucidating the diffusion mode of a fluorescent protein (FP) in the endoplasmic reticulum (ER) lumen of living cells.

## Results and Discussion

### SpeedyTrack records microsecond dynamics by spreading wide-field single-molecule images into streaks

CCD works by shifting the photo-generated charge carriers along the light-sensing element array for serial readout. Modern EM-CCDs often use a rectangular CCD chip that is vertically divided into equally sized active and storage areas, *e.g.*, each of 512 × 512 pixels. The active area on the top is exposed for signal, whereas the storage area at the bottom is masked. For continuous recording in the typical frame-transfer mode, after each exposure, the image of the entire active area is rapidly shifted down to the storage area at 0.3–0.5 μs/row in ~200 μs, where all pixels are serially read out as the active area is exposed for the next image. Since all pixels in the storage area need to be read out before the next exposed image can be transferred in, the achievable framerate is limited by readout to typically ~18 and ~9 ms per frame for frame sizes of 512 and 256 rows.

We ask whether it is possible to leverage the native sub-microsecond vertical shifting speed of CCDs to record fast single-molecule tracks in the wide field. In the known "fast kinetic mode", a very small frame size (*e.g.*, height $h = 6$ rows) of the CCD chip is exposed, so that after each exposure, charges are shifted down by the full image height $h$ to form an image array on the chip. A series of fast images is thus collected for the extremely small field of view[18,19].

We reason that for sparse molecules in the wide field (image height $h > 100$ rows), the number of rows shifted after each exposure can be much smaller than the height of the exposed image $h$. Instead, one only needs to stagger out the single-molecule images at different time points, for which a $\delta \sim 10$-row vertical shift suffices to account for the point spread function (PSF) and molecular motion. This shift can be accomplished in ~5 μs after each ~100 μs exposure, so that repeated rounds of exposure-vertical shift (Fig. 1a) generate a trail (streak) of images for each molecule, with each image corresponding to one discrete timepoint (Fig. 1b). With a typical EM-CCD chip of 1024 rows, the 10-row shift scheme allows a molecule to be recorded for up to 100 timepoints along the vertical direction of the chip. The time domain information of single-molecule images is thus projected to the vertical spatial domain (Fig. 1b). After readout, each trail of single-molecule images is separately processed, in which molecular localizations are subtracted by the preset vertical shifts to collapse into spatial trajectories (Fig. 1c).

Whereas spatial encoding of time-domain information has been previously achieved for immobilized single molecules down to 4-ms time resolution based on mechanical scanning[20,21], the SpeedyTrack scheme utilizes the native vertical shifting capability of the CCD, and so may be performed on any EM-CCD-based single-molecule microscope without modifications to existing optics or electronics, while retaining the high sensitivity of EM-CCD. We have experimentally demonstrated SpeedyTrack with two commercial EM-CCDs in hand (Supplementary Fig. 1), and we provide our codes online (Code Availability). We expect SpeedyTrack to be further compatible with other EM-CCDs with minimal software configurations.

SpeedyTrack is uniquely tailored for single-molecule/single-particle imaging, so that point-like single-molecule images in the wide field are clearly resolved after being spread vertically in $\delta \sim 10$-row increments (Fig. 1b). For continuous structures, ~10-row shifts in the wide field lead to unresolvable signal overlap (Fig. 1d). This is

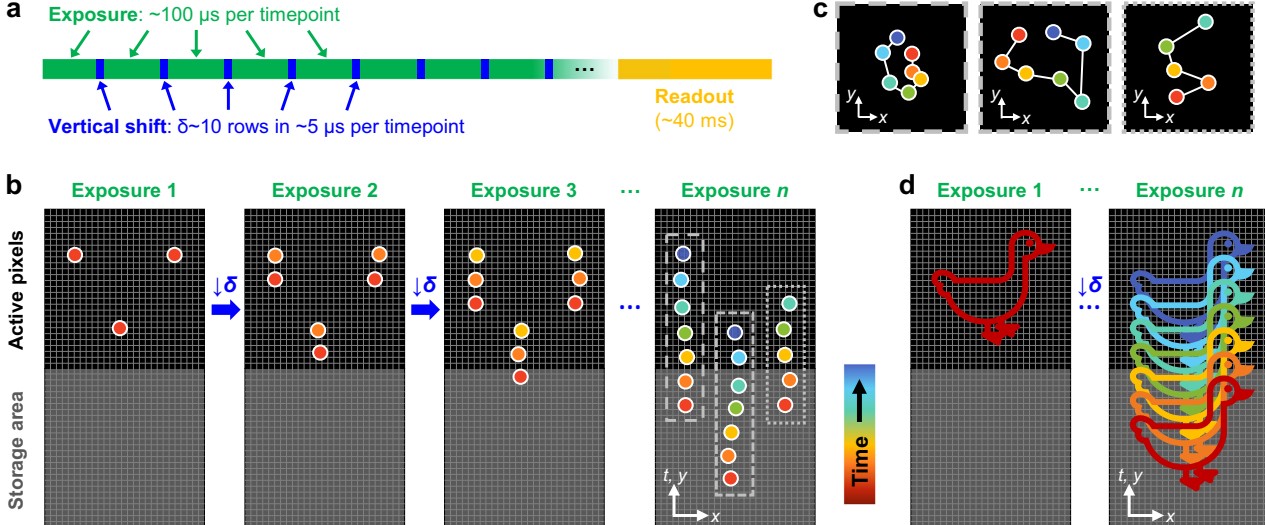

**Fig. 1 | Schematics: SpeedyTrack leverages the native fast vertical shifting of EM-CCD to record microsecond single-molecule dynamics in the wide field. a** CCD timing diagram of SpeedyTrack. Many rounds of exposure-vertical shift are made before the final readout. **b** SpeedyTrack recording of single-molecule trajectories. After each exposure, the entire frame is shifted down by δ ~ 10 rows, just enough to separate the single-molecule images at each timepoint. A trail (streak) of images (colored by timepoint) is thus generated for each molecule along the vertical direction of the CCD chip. After a series of exposures, the chip-stored image is read out collectively, which contains multiple single-molecule streaks projecting discrete timepoints in the vertical direction. **c** Single-molecule trajectories reconstructed from the three streaks depicted in (**b**) by deducting the preset vertical shifts between timepoints. **d** The SpeedyTrack exposure-shift scheme is tailored for single-molecule/single-particle imaging: for continuous structures, ~10-row shifts in the wide field lead to unresolvable signal overlap.

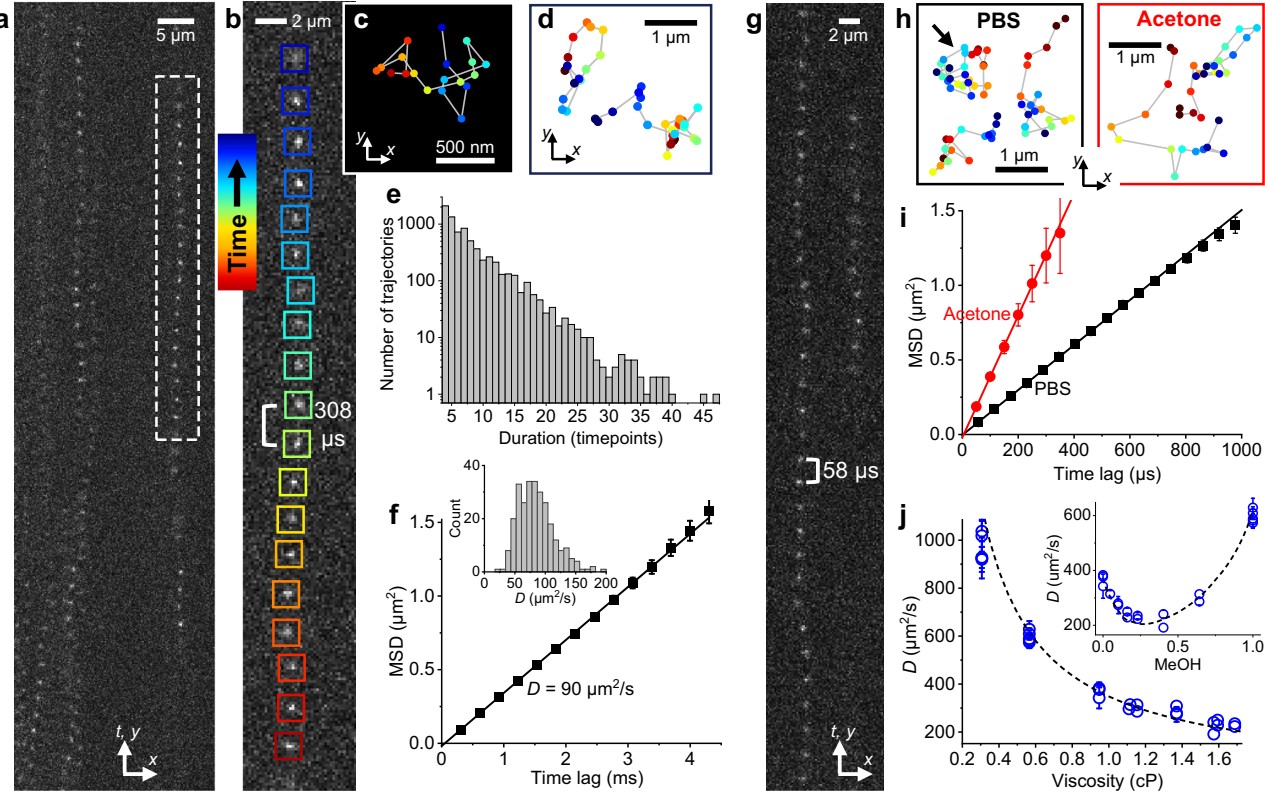

**Fig. 2 | Fast wide-field tracking of freely diffusing molecules through Speedy-Track. a–f** SpeedyTrack data of Cy3B-labeled carbonic anhydrase diffusing in PBS, with an exposure time of 300 μs and a vertical shift time of 7.5 μs for 15 rows at each timepoint. **a** A portion of one CCD frame, showing multiple streaks from single molecules entering/exiting the focal plane at different locations and timepoints. **b** Zoom-in of one streak, with colored boxes indicating different time points. **c** Single-molecule trajectory reconstructed from the localizations in (**b**) by back-shifting 15 rows between consecutive timepoints, colored as the boxes in (**b**). **d** Additional example trajectories. **e** Distribution of durations for single-molecule trajectories collected over 4000 SpeedyTrack frames. **f** Squares: MSD at different time lags calculated from the trajectories. Error bars: standard errors. Line: Linear fit to the first 4 data points, yielding a diffusion coefficient of $90 \pm 3\,\mu m^2/s$ (95% confidence interval) from 5573 tracks. **Inset**: distribution of apparent diffusion coefficients estimated for individual trajectories longer than 12 timesteps. **g–j** SpeedyTrack data of Cy3B free dye. **g** A portion of one SpeedyTrack frame showing

two single-molecule streaks for Cy3B diffusing in PBS, with exposure and vertical-shift times of 50 and 7.5 μs at each timepoint. **h** Examples of reconstructed Cy3B trajectories in PBS and acetone at 57.5 and 50 μs timesteps, respectively. The arrow points to the trajectory reconstructed from the long streak in (**g**). **i** Data points: MSD vs. time lag for Cy3B in PBS (black, 1860 tracks) and acetone (red, 1924 tracks). Error bars: standard errors. Lines: Linear fits to the first 4 data points, yielding diffusion coefficients of $378 \pm 9$ and $1022 \pm 30;\mu m^2/s$ (95% confidence intervals), respectively. **j** Circles: SpeedyTrack-measured diffusion coefficients of Cy3B in solvents and mixtures of varying viscosities. Each data point is obtained from the MSD-time lag fit to one experiment of $10^3 - 10^4$ tracks (Supplemental Table 1). Error bars: 95% confidence intervals. Dashed curve: Trend predicted by scaling the PBS value by the medium viscosity. **Inset**: Results in methanol-water mixtures of varying methanol mole fractions, compared to the expected trend based on the known mixture viscosity.

reminiscent of our previous development of spectrally resolved SMLM[22,23], wherein spectral dispersion in the wide field yields resolvable spectra for point sources but not for continuous structures[24].

### Fast wide-field tracking of freely diffusing molecules

We first demonstrate SpeedyTrack for tracking molecules freely diffusing in the wide field. A ~100 pM fluorescent sample was illuminated at an incidence angle slightly lower than the critical angle of the coverslip-sample interface. The focal plane was held ~3 μm into the sample for the wide-field recording of single molecules that diffused into the focal plane. For SpeedyTrack, for each timepoint we exposed for a fixed duration of 40–300 μs as needed (below) and then shifted the entire image down by $\delta = 15$ rows at 0.5 μs/row in 7.5 μs. This exposure-shift scheme was repeated 68 times to fill the CCD chip with single-molecule streaks; the signal stored on the CCD chip was then collectively read out in ~40 ms as one SpeedyTrack frame, and the next acquisition cycle started again.

Figure 2a-f presents SpeedyTrack data on Cy3B-labeled carbonic anhydrase (30 kDa) diffusing in phosphate-buffered saline (PBS). The

exposure time was set as 300 μs, which together with the 7.5 μs shift time per timepoint yielded a temporal resolution of 307.5 μs. As individual molecules stochastically diffused in and out of the focal plane, they each created a streak of images in the readout frame (Fig. 2ab and Supplementary Video 1), with the streak length corresponding to the number of timepoints captured. The wide-field detection format allowed us to track molecules entering anywhere in the field of view, while diffusion dynamically replenished molecules to overcome photobleaching. We thus were able to repeat SpeedyTrack thousands of times to sample many molecules at a relatively constant molecular density in the view (Supplementary Video 1). The typical single-molecule image density in the acquired data was ~100 per frame, and an upper limit of ~500 single-molecule images per frame is recommended to avoid overlapping of trajectories.

Single-molecule images were localized and grouped into trails using custom codes (Methods and Code Availability). For each trail, localizations were collapsed into spatial trajectories by deducting the known vertical shift ($\delta = 15$ rows) between timepoints (Fig. 2b-d). The resultant trajectories, with typical durations of ~4–40 timepoints following an exponential distribution likely limited by diffusion out of the

focal plane (Fig. 2e and Supplementary Fig. 2), can then be processed as in conventional SPT, *e.g.*, analyzed for mean-squared displacements (MSDs) at different time lags $t_{lag}$ (Fig. 2f). A linear MSD-$t_{lag}$ dependence was observed, from the slope of which a diffusion coefficient $D$ of 90 μm²/s was determined based on MSD = $4Dt_{lag} + b$. While SM*d*M reports comparable $D$ values[25] based on simple statistics of displacements at a single timestep, SpeedyTrack trajectories enabled MSD-$t_{lag}$ analysis to confirm a normal diffusion model and remove uncertainties in $D$ quantification due to single-molecule localization uncertainties (static error) and motion blur (dynamic error), which oppositely shift MSD-$t_{lag}$ plots toward positive and negative intercepts $b$, respectively[26,27]. By obtaining long trajectories, an apparent $D$ value can be further assessed for each molecule (Fig. 2f inset), which is impossible with SM*d*M. This capability is powerful for resolving different molecular states, as demonstrated below for DNA strands in combination with smFRET.

We next explore the more challenging case of free dye diffusion. Figure 2g shows example SpeedyTrack streaks for the 560 Da dye Cy3B diffusing in PBS. To capture fast motion, the exposure time was reduced to 50 μs, which together with the 7.5 μs shift time gave a temporal resolution of 57.5 μs. SpeedyTrack successfully recorded single-molecule trajectories for > 30 timepoints (Fig. 2gh). MSD-$t_{lag}$ analysis (Fig. 2i) extracted a diffusion coefficient of 378 μm²/s, consistent with expected values[28–30].

For even faster diffusion, we examined Cy3B in acetone. At a temporal resolution of 50 μs, SpeedyTrack determined a diffusion coefficient of 1022 μm²/s through MSD-$t_{lag}$ analysis (Fig. 2hi), consistent with the ~32% viscosity of acetone versus water at room temperature. Comparing the SpeedyTrack-determined diffusion coefficients in different solvents and mixtures showed good agreement with that expected from the medium viscosity over a wide range (Fig. 2j and inset), testifying to SpeedyTrack's accuracy.

As noted, wide-field SPT is often limited to slow, bound diffusion. While SM*d*M has been pushed to a 400-μs timestep across paired frames to detect fast diffusion[31], further improvement is unlikely given the typical ~200 μs exposure dead time between frames dedicated to frame transfer. Although the EM-CCD "fast kinetic mode" accesses sub-ms time resolutions, the need to shift the height of the entire field of view after each exposure seriously compromises between the achievable speed, image size, and track length, and such approaches have not been demonstrated for the SPT of freely diffusing molecules. In contrast, by shifting only ~10 rows per exposure, SpeedyTrack readily achieves 50 μs resolution and uniquely enables the wide-field recording of single-molecule trajectories under free, fast diffusion. It is further worth mentioning that SpeedyTrack does not require stroboscopic illumination or laser-camera synchronization, major instrumental challenges of SM*d*M. As a limitation, SpeedyTrack requires sparse single molecules to avoid overlapping of the stretched single-molecule image trails.

While the increased temporal resolution of SpeedyTrack limits the detected photons per localization, for Cy3B we were able to collect >100 photons in 50 μs to achieve a localization precision of ~30 nm (Supplementary Fig. 3). Longer exposure times and/or higher excitation powers yielded higher photon counts and precision, and the measured localization errors at different photon counts and background levels agreed with theoretical predications[32] after accounting for the EM-CCD multiplicative noise (Supplementary Fig. 3). Use of brighter fluorophores (e.g., nanoparticles) should allow higher spatial resolution. The repeated vertical shifts employed by SpeedyTrack accumulate background noise from the multiple exposures in the frame and thus reduce the achievable localization precision (Supplementary Fig. 4); the use of larger vertical shifts alleviates this issue at the expense of a decreased maximum track length (Supplementary Fig. 4).

## Tracking sub-ms FRET dynamics and resolving states for freely diffusing single molecules

We next integrate SpeedyTrack with wide-field smFRET to track the conformational dynamics and interaction states of freely diffusing molecules. The sample was tagged with a Cy3B-Atto 647 N donor-acceptor dye pair, and the donor was excited. A dichroic mirror separated the donor and acceptor wide-field emissions for projecting onto two different areas of an EM-CCD[16,17] that ran in the Speedy-Track mode.

Figure 3a and Supplementary Fig. 5 present example SpeedyTrack single-molecule streaks in a Tris buffer for the free diffusion of a dual-labeled DNA hairpin that gives high and low FRET signals for its closed and open states, respectively (Fig. 3b). FRET efficiency was calculated for each timepoint using the detected single-molecule intensities in the donor and acceptor channels, thus capturing switching between the two states at 0.8- or 0.3-ms resolutions for the freely diffusing hairpin molecules (Fig. 3c and Supplementary Fig. 5). From thousands of such smFRET time traces, we calculated the time-correlated conditional probability for a molecule to stay in the high-FRET (closed) state as $P_{cc}(t_{lag}) = P(\text{closed}, t+t_{lag} \mid \text{closed}, t)$ for varied $t_{lag}$ (Fig. 3e)[33], assuming negligible photobleaching over the ~10 ms observation window given the 110 ms photobleaching time constant (Supplementary Fig. 2). Exponential decays toward equilibrium were observed, as expected. Fitting these time dependencies (Methods) yielded decreasing and increasing rates for hairpin opening and closing, respectively, for experiments ran under increasing salt (Fig. 3f), consistent with that expected from the ionic screening of DNA charges[34]. SpeedyTrack-smFRET thus elucidated conformational dynamics for freely diffusing DNA hairpin molecules. MSD-$t_{lag}$ analysis of the SpeedyTrack-smFRET data at both 0.8- and 0.3-ms time resolutions further yielded comparable diffusion coefficients of ~58 μm²/s for the freely diffusing hairpin (Supplementary Fig. 5), as expected for a DNA construct of this size[35].

A unique feature of SpeedyTrack is that for every molecule, its diffusion trajectory is reconstructed together with its FRET time trace (*e.g.*, Fig. 3d). To show how this capability may help resolve single-molecule states, we examined the hybridization of a donor dye-tagged short (8-base) DNA strand with an acceptor dye-tagged complementary strand 100-base in length (Fig. 3g). SpeedyTrack was performed at 300 μs exposure time and 7.5 μs shift time per timepoint, and trajectories longer than 12 timepoints were each calculated for a mean FRET value and assigned an estimated diffusion coefficient[36]. The two-dimensional distribution of these two single-molecule parameters (Fig. 3h) showed two major populations: The high FRET and low diffusivity state corresponded to the hybridized DNA, whereas the low FRET and high diffusivity state corresponded to the unhybridized short DNA (Fig. 3h insets). A small population with low FRET and low diffusivity was attributed to hybridized DNA lacking an acceptor dye. Repeating the measurement with a 50 μs exposure time yielded similar features (Supplementary Fig. 6). SpeedyTrack-smFRET thus resolved different single-molecule states at high temporal resolution.

## Super-resolution mapping of fast single-molecule trajectories via time-encoded variable vertical shifts

While SpeedyTrack enables high-speed single-molecule tracking in the wide field, the streaks spread across the CCD chip convolve vertical position and time. This confusion is unproblematic for probing solution-phase diffusion and dynamics, as each single-molecule streak is converted into a trajectory relative to the first timepoint. For applications in which spatial patterns are important, *e.g.*, intracellular diffusion, deconvolution of the spatial and temporal dimensions becomes necessary.

To deconvolve position and time without incurring additional optics, we introduce a temporally patterned vertical shifting scheme to encode time. After each exposure, the wide-field image is shifted down

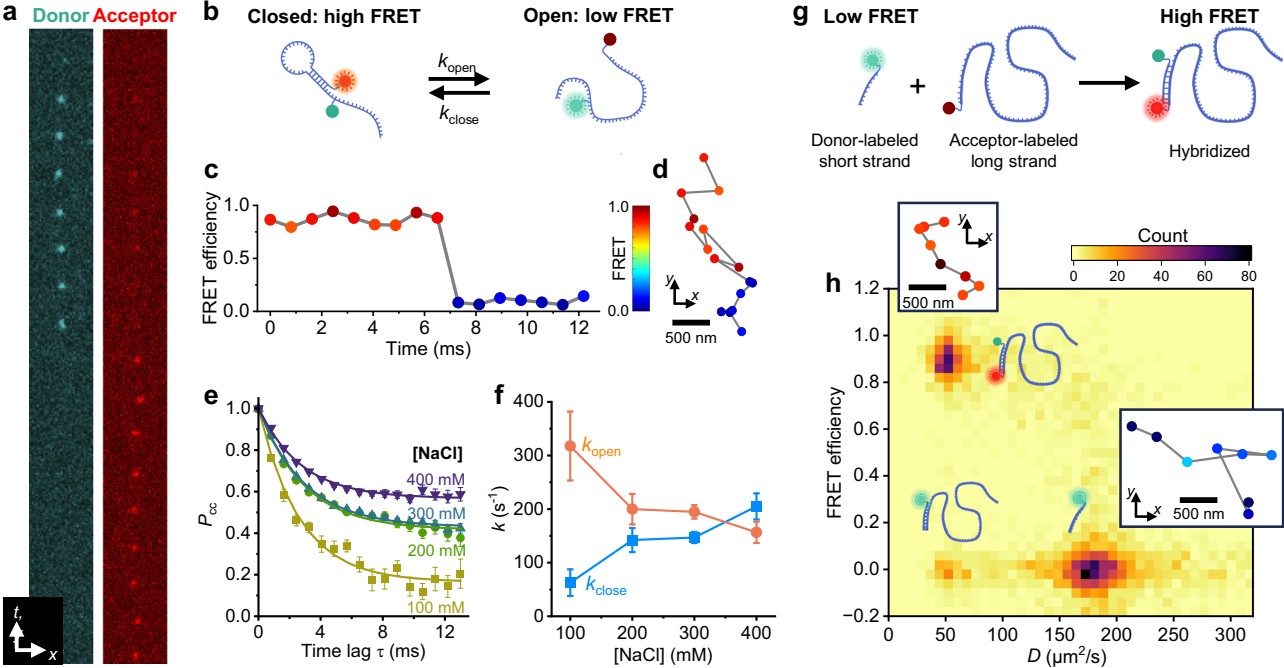

**Fig. 3 | Integration of SpeedyTrack and smFRET for freely diffusing molecules.**
**a–f** Results with a donor-acceptor dual-labeled DNA hairpin. For each timepoint, the exposure time was 800 µs and the vertical shift time was 12.5 µs for 25 rows. **a** An example SpeedyTrack streak in the donor and acceptor channels for a hairpin molecule freely diffusing in a Tris buffer (200 mM NaCl). **b** Schematic: Closing and opening of the hairpin give high and low FRET signals, respectively. **c** FRET efficiency time trace from (**a**), calculated for each timepoint as the detected single-molecule intensity in the acceptor channel divided by the sum intensity of both channels. **d** Reconstructed spatial trajectory, colored by the FRET value. **e** Time-correlated conditional probabilities for a molecule to stay in the high-FRET (closed) state in solutions containing different amounts of NaCl, generated from ~5000 SpeedyTrack smFRET time traces under each condition (Supplemental Table 2).

Error bars: standard errors via bootstrapping. **f** Hairpin opening and closing rates, $k_{open}$ and $k_{close}$, extracted from the correlation results in (**e**) by fitting to an exponential decay model (Methods). Error bars: 95% confidence intervals. **g–h** Results on the hybridization of a donor dye-tagged 8-base DNA strand with an acceptor dye-tagged complementary 100-base strand. Exposure time was 300 µs and vertical shift time was 7.5 µs for 15 rows for each timepoint. **g** Schematic: The short strand and the hybridized strand give low and high FRET signals, respectively. **h** Two-dimensional distribution of mean FRET efficiency vs. estimated diffusion coefficient for individual single-molecule trajectories longer than 12 timepoints. **Insets**: Schematics of the three resolved states, and example trajectories (307.5 µs time-steps) colored by the FRET value as in (**d**).

by varied numbers of rows, *e.g.*, different multiples (*m*) of a base value like 15, according to a predefined pattern (Fig. 4a). This approach, variable shifting (VS)-SpeedyTrack, creates time-dependent gaps between the recorded single-molecule images, which can be used as a barcode to align each streak with the master sequence (Fig. 4b) to recover time and deconvolve absolute position (Fig. 4c).

To experimentally demonstrate VS-SpeedyTrack with distinct spatial patterns, we examined the diffusion of Dendra2, a 26 kDa photoconvertible FP, in the ER lumen of living COS-7 cells. When SM*d*M was recently applied to this system, normal diffusion had to be assumed to make use of the single-molecule displacements detected at a fixed 1-ms timestep[37]. However, an earlier study suggested flow-facilitated intra-ER transport[38], demanding further assessment of the diffusion mode that may be uniquely elucidated by trajectory analysis through VS-SpeedyTrack.

VS-SpeedyTrack was performed at 500 µs time resolution, under which condition ~150 photons were detected from each single-molecule image to achieve a localization precision of ~25 nm. The vertical-shift pattern shifted *m* = 1–4 multiples of 15 rows after each exposure to accommodate a total of 33 exposures, and was designed so that any subsequence of 4 exposures or longer is unique within the sequence (Methods). A weak 405 nm laser photoconverted a very small fraction of Dendra2 into the red state to be excited. The applied power of this photoconversion laser was gradually adjusted during the experiment so that emitting molecules were observed at a controlled density (~120 single-molecule images/frame) for > 50,000 Speedy-Track frames (Supplementary Fig. 7 and Supplementary Video 2).

Single-molecule trails 5–28 timepoints in duration were extracted, and their shift patterns were aligned with the master sequence (Supplementary Fig. 7). As any sequence of 4 exposures is distinct, the alignment was robust against moderate mismatches (Methods), and Dendra2 exhibits minimal single-molecule blinking at the (sub)ms time scales of our measurements[39]. Matched trails were assigned a time in the master sequence, based on which absolute *y*-positions were recovered by deducting the patterned vertical shifts in the sequence to generate trajectories in real space (Methods).

We first pooled all single-molecule localizations from the deconvolved real-space trajectories to generate an SMLM image. The resultant image (Fig. 4d) agreed with the diffraction-limited epifluorescence image (Fig. 4e), while further resolving nanoscale ER tubules[40] not visible in the latter. Thus, we have correctly deconvolved molecular positions.

As the single-molecule localizations above were from trajectories of 5 or more timepoints, the observed continuous SMLM images indicate that the VS-SpeedyTrack-collected trajectories have traversed the ER lumen space. Overlaying a fraction of the trajectories over the SMLM image visualized how molecules navigated through the ER network at 500 µs time resolution (Fig. 4f), signifying an opportunity to systematically elucidate diffusion patterns through SPT analysis.

We start by analyzing the pooled single-molecule trajectories in a straight ER-tubule segment (Fig. 4g inset). Given the high geometric aspect ratio, MSD values were decomposed into two components parallel and perpendicular to the tubule's running direction. Plotting these two components versus $t_{lag}$ (Fig. 4g) showed strong confinement

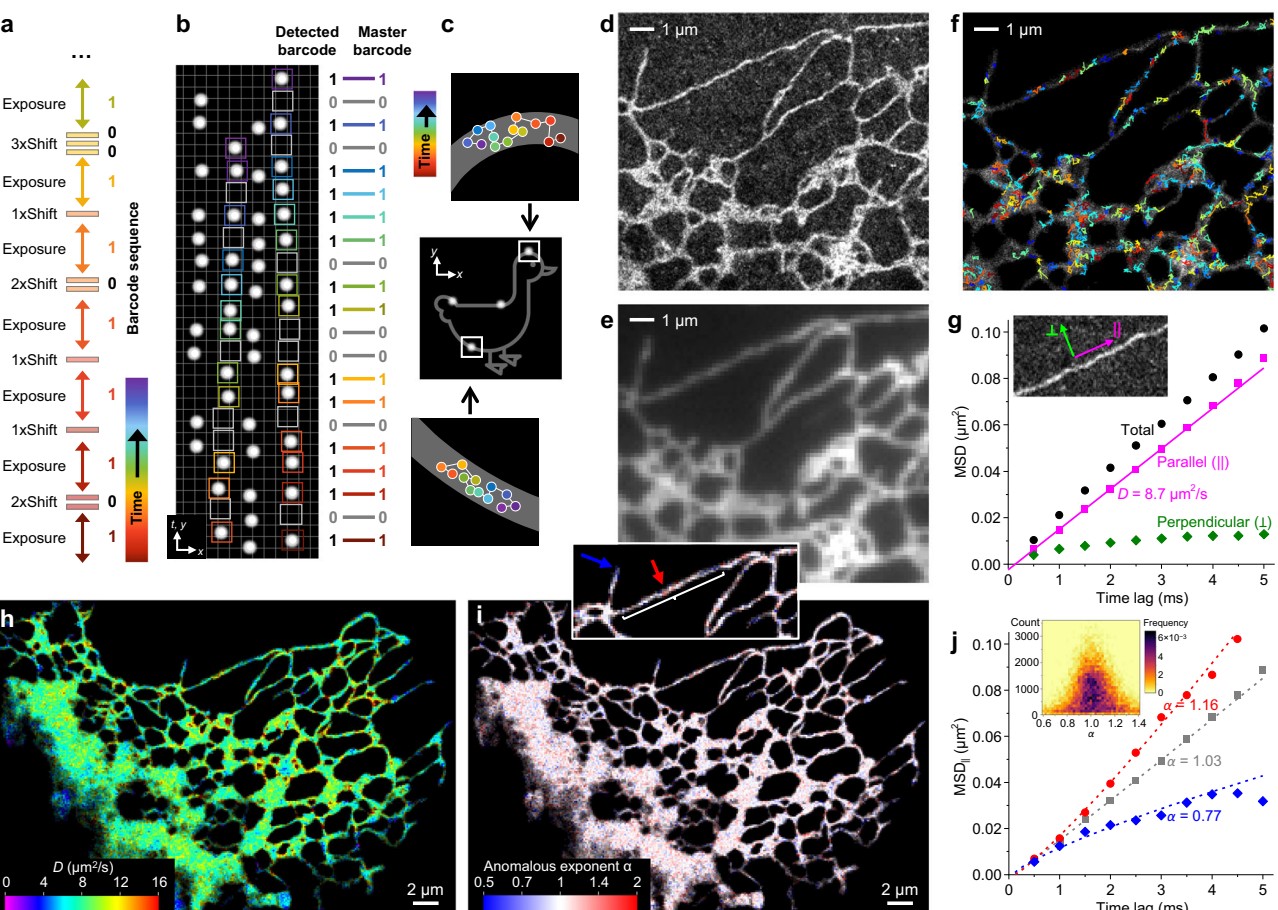

**Fig. 4 | VS-SpeedyTrack deconvolves temporal and spatial information to achieve super-resolution mapping of fast single-molecule trajectories. a–c** Schematics. **a** Example time-encoded shift scheme: After each exposure, the entire frame is shifted down by varied numbers of rows, here as m = 2, 1, 1, 2, 1, and 3 multiples of a base value. **b** Resultant VS-SpeedyTrack readout, showing four single-molecule streaks with varied gap patterns due to time-encoded shifts. Example alignments with the master sequence are highlighted for two streaks using matched-color boxes and with the barcode written out for the rightmost streak. **c** Deconvolved real-space trajectories of the four molecules in (b), with those from the two color-boxed streaks shown as zoom-ins. **d–j** Experimental results of Dendra2 FP diffusing in the ER lumen of a living COS-7 cell, performed at 500 µs resolution. **d** SMLM image constructed from all single-molecule localizations from all deconvolved trajectories. Similar results were repeated three times. **e** Diffraction-limited epifluorescence image of the same region. **f** Overlay of

example trajectories over the SMLM image, colored by their estimated D. **g** Data points: MSD values (black) at different time lags for all trajectories in a straight ER-tubule segment (**inset**), and their decomposition into components parallel (∥, magenta) and perpendicular (⊥, green) to the tubule's running direction. Line: linear fit to the first four data points of $MSD_\parallel$, yielding D = 8.7 µm²/s from the slope under a one-dimensional diffusion model. **h** Color-coded D map constructed by calculating intra-ER trajectories into single-molecule displacements at varied time lags and binning them onto a 120 nm grid, determining the local principal direction of each spatial bin, and then fitting the $MSD_\parallel$-$t_{lag}$ dependence as in (**g**). **i** Map of anomalous exponent α obtained by the power-law fitting of local $MSD_\parallel$-$t_{lag}$ dependence to $MSD_\parallel = 2Kt_{lag}^\alpha + b$. **j** The power-law fitting results for the straight ER-tubule segment in (**g**) (gray) and for two regions of low and high α values pointed to by the blue and red arrows in the inset of (**i**). **Inset**: Two-dimensional distribution of α vs. count of displacements, for a 160 nm bin size.

for the latter ($MSD_\perp$), which stayed flat except for a minor initial increase, expected given the ~100 nm tubule lumen width[41]. Meanwhile, the parallel component ($MSD_\parallel$) increased linearly over $t_{lag}$ (Fig. 4g), suggesting normal diffusion. A linear fit to the four initial $MSD_\parallel$ values with a one-dimensional diffusion mode $MSD_\parallel = 2Dt_{lag} + b$ yielded D = 8.6 µm²/s, consistent with previously reported intra-ER FP diffusivities[37,42–44].

For spatial mapping, we calculated intra-ER trajectories into single-molecule displacements at varied time lags of 1–10 timesteps, and then binned these time-varied displacements onto a 120 nm spatial grid for local statistics. For each spatial bin, a principal diffusion direction was determined[45], based on which local MSD values were decomposed into parallel and perpendicular components. Linear fits of the resultant $MSD_\parallel$-$t_{lag}$ relationship in each bin, like that done above for the tubule segment, thus generated a map of local D (Fig. 4h), which appeared largely homogenous throughout the cell.

The many single-molecule trajectories obtained by SpeedyTrack at 500 µs resolution further uniquely allowed us to examine how well molecular motion conforms to normal diffusion. To this end, for the time-varied displacements sampled by each 120 nm × 120 nm bin, we fit $MSD_\parallel$ of the first 6 time lags to the power law $MSD_\parallel = 2Kt_{lag}^\alpha + b$ (Methods). The resultant anomalous exponent α, which was rendered into a color map (Fig. 4i), centered around 1. A fit to the $MSD_\parallel$-$t_{lag}$ relationship of the straight tubule segment examined in Fig. 4g yielded α = 1.03 (gray in Fig. 4j), whereas lowered α values are often observed at tubule ends (blue in Fig. 4i inset and Fig. 4j), expected from their local diffusion confinements. We also examined the occasionally observed locally elevated α, e.g., red arrow in Fig. 4i inset, where accelerated $MSD_\parallel$ over $t_{lag}$ was fit to α = 1.18 (red in Fig. 4j). To understand whether such local observations could be due to statistical fluctuations, we plotted α vs. count of displacements in each bin as a two-dimensional heatmap, in this case for a larger bin size of 160 nm × 160 nm (Fig. 4j

inset). It is thus found that for bins with varying displacement counts, increased counts converged $\alpha$ to 1. Simulation recapitulated similar count-based convergences for molecules diffusing under normal diffusion (Supplementary Fig. 8), suggesting that statistical fluctuations, rather than active processes, led to the occasionally observed local $\alpha > 1$ behavior. Together, our results indicate that passive normal diffusion is the predominant transport mechanism in the ER lumen. This conclusion is in line with that recently deduced from diffraction-limited photoactivation imaging at the whole-cell level[46]; yet, VS-SpeedyTrack mapped out the diffusion mode for the ER network at millisecond-nanometer scales.

Taking advantage of the native sub-microsecond vertical shifting capability of common EM-CCDs, SpeedyTrack achieves fast wide-field single-molecule tracking/imaging by staggering self-confined, PSF-sized single-molecule images along the CCD chip at ~10-row spacings between consecutive timepoints, thus effectively projecting the time domain to the vertical spatial domain. Consequently, SpeedyTrack tracked molecules freely diffusing in the wide field at temporal resolutions down to 50 μs. The resultant single-molecule trajectories, up to 40 timepoints in duration, uniquely enabled SPT analysis for freely diffusing molecules, confirming normal diffusion modes and quantifying diffusion coefficients up to 1000 μm²/s. Integration with wide-field smFRET to concurrently acquire diffusion trajectories and FRET time traces next elucidated the conformational dynamics and binding states of freely diffusing molecules. Although SpeedyTrack confuses temporal and spatial information in its basic form, we developed a temporally patterned vertical shifting scheme to deconvolve the two and map fast single-molecule trajectories at the super-resolution level, thus resolving the diffusion mode of an FP in the ER lumen with nanoscale mapping of local MSD-$t_{lag}$ behavior. While many of the above-demonstrated capabilities substantially outperform existing approaches, SpeedyTrack further stands out for its simplicity by directly working off the built-in vertical shift of common EM-CCDs without the need to modify existing optics or electronics. We thus provide a facile solution to the microsecond tracking of single molecules and their super-resolution mapping in the wide field.

## Methods

### Preparation of in vitro samples

Carbonic anhydrase (Sigma C2624) was incubated with a 10-fold molar excess of Cy3B-NHS ester (Cytiva PA63101) in 0.1 M NaHCO₃ at room temperature for 2 h before being filtered through an Amicon centrifugal filter (30 kDa MWCO) to remove unlabeled dye. Absorption spectroscopy through Nanodrop (Thermo Fisher) indicated a 0.9 dye-to-protein ratio for the product. Cy3B free dye (carboxylic acid) was from Lumiprobe (2321).

DNA hairpin for smFRET was synthesized with the acceptor dye Atto 647 N at the 5′ and contained a 5-octadiynyl dU handle for click chemistry labeling of the donor dye (IDT, sequence: Atto 647N-TGG GTT AAA AAA AAA AAA AAA AAA AAA AAA AAA AAA CCC ATT TCT TCA C 5-octadiynyl dU A ACC AGT CCA AAC TAT CAC AAA CTT A-biotin). 17 μM of this DNA hairpin was mixed with 50 μM Cy3B azide (Lumiprobe 19330), 0.5 mM tris(benzyltriazolylmethyl)amine (TBTA), and 0.5 mM ascorbic acid in 30 μL copper(II)-TBTA catalytic buffer (Lumiprobe 61150) and allowed to react overnight. The labeled DNA hairpin was precipitated and washed with cold ethanol, and then resuspended and further purified by an ÄKTA pure chromatography system (Cytiva) using a Superdex 75 Increase 3.2/300 column.

Samples for DNA hybridization experiments were prepared by NHS-ester reaction with amine-modified DNA strands (IDT, short oligo sequence: C6 amino- CGC CCG GG, long oligo sequence: C6 amino-TCC CGG GCG TTT TTT TTT TTT TTT TTT TTT TTT TTT TTT TTT TTT TTT TTT TTT TTT TTT TTT TTT TTT TTT TTT TTT TTT TTT TTT TTT TTT TTT TTT T). ~0.15 mM oligos in 0.1 M sodium bicarbonate were mixed with a 10-fold molar excess of dye NHS esters (Cy3B and Atto

647 N for the short and long oligos, respectively). Oligos were purified by ÄKTA. Nanodrop indicated ~50% of the short oligos to be labeled with Cy3B and ~75% of the long oligos to be labeled with Atto 647 N.

In typical experiments, the fluorescently labeled sample was added to ~500 μL buffer or solvent at ~100 pM. Protein samples used Dulbecco's phosphate-buffered saline. DNA samples used Tris-EDTA buffers with the addition of 100–400 mM NaCl, 160 units/mL glucose oxidase, and 2600 units/mL catalase. DNA hairpin sample buffers were supplemented with 2 mM Trolox and 5% glucose. DNA hybridization sample buffers were supplemented with 2 mM ascorbic acid, 2 mM methyl viologen, and 10% glucose. Protein and DNA samples were imaged in chambered coverslip devices with a non-adherent Bioinert surface (ibidi 80800). Free dye experiments were performed with cleaned 25 mm coverslips mounted with Attofluor cell chambers (Invitrogen A7816).

### Preparation of cell samples

COS-7 cells (University of California Berkeley Cell Culture Facility) were cultured in phenol-red-free DMEM with 10% fetal bovine serum, non-essential amino acids (NEAA), Glutamax, and penicillin/streptomycin at 37 °C and 5% CO₂. Cells were seeded onto cleaned 18 mm glass coverslips at ~10% confluency 48–72 h prior to imaging. 24 h after plating, cells were transfected using Lipofectamine 3000 with 1 μg ER-Dendra2 plasmid (a gift from Michael Davidson, Addgene plasmid #57716). Prior to imaging, cells were washed with the imaging medium (L-15, Gibco 21083027) and transferred to coverslip holders.

### Microscope setup

SpeedyTrack was performed on two typical SMLM systems based on Nikon Ti-E inverted fluorescence microscopes[14,47]. Briefly, a 561 nm excitation laser and a 405 nm activation laser (for photoactivation of Dendra2) were focused at the edge of the back focal plane of an oil immersion objective lens (CFI Plan Apochromat Lambda 100×, NA = 1.45), so that light entered the sample at an angle slightly below the critical angle of total internal reflection at the coverslip-sample interface, an approach commonly known as pseudoTIRF or HILO[48,49]. The sample was thus illuminated and imaged in the wide field ~3 μm above the coverslip surface for molecules that stochastically diffused into the focal plane. Fluorescence emission was detected with an EM-CCD. A standard Andor iXon Ultra 897 EM-CCD was used for most of the presented data, except in Supplementary Fig. 1, where we further compare results with a Nuvu HNü 512 EM-CCD. Both EM-CCDs are based on a rectangular CCD chip of 512 × 1024 pixels that is vertically divided into equally sized active (exposed) and storage (masked) areas each of 512 × 512 pixels. Our illumination covered a ~ 35 μm sample area, which corresponded to ~220 rows of pixels on the camera. For smFRET, wide-field emission was cropped by a rectangular aperture at the image plane and then split into donor and acceptor channels using a dichroic mirror (T647lpxr-UF3, Chroma) placed between a set of relay lenses, before being refocused onto two different areas of the EM-CCD. All measurements were performed at room temperature (23 ± 0.5° C).

### SpeedyTrack of in vitro samples

SpeedyTrack was performed using custom Python codes (Code Availability) that invoked standard API functions of the EM-CCD. For SpeedyTrack of in vitro samples with fixed vertical shifting, the EM-CCD ran under the internal trigger mode. The exposure time for each timepoint was set to a fixed value, typically 40–300 μs for fast diffusing molecules and 300–800 μs for integration with smFRET of DNA. The number of rows to shift down after each exposure ("shift height" $\delta$) was set to a fixed value, with 15 being typical (tested range 10–25), to comfortably stagger out single-molecule images at successive timepoints. With a typical 2 MHz vertical clock (0.5 μs/row), the 15-row vertical shift took 7.5 μs. The exposure-shift scheme was typically

repeated 68 times to fill the 512 × 1024 CCD chip with single-molecule streaks. The signal stored on the CCD chip was then read out as one SpeedyTrack frame with an EM gain of 300. At a 17 MHz readout rate, reading out the full frame took ~40 ms, after which the next Speedy-Track acquisition cycle started again. A typical run acquired 1000-10,000 SpeedyTrack frames, during which molecules dynamically diffused in and out of the focal plane to maintain a relatively constant molecular density in the collected data. The typical single-molecule image density in the acquired SpeedyTrack data, which was readily tuned during the experiment by titrating the sample concentration, was ~100 per frame, and an upper limit of ~500 single-molecule images per frame is recommended to avoid overlapping of trajectories.

## VS-SpeedyTrack of ER-Dendra2 in live cells

For VS-SpeedyTrack of ER-Dendra2 in live cells, the EM-CCD was set to an external trigger mode, in which the CCD chip was continuously exposed but responded to every trigger event (voltage rising edge) by vertically shifting down the frame 15 rows in 7.5 μs. The external trigger was applied using an arbitrary waveform generator (AWG; Siglent SDG1025) that was loaded with a predefined shifting scheme. The sequence had 33 exposures each followed by 1–4 voltage pulses of 9 μs duration to trigger the CCD chip to execute 1–4 vertical shifts of 15 rows, while maintaining a fixed total time of 500 μs for each timestep. The sequence, noted as the number of voltage pulses after each exposure (and hence $m$, the vertical shifts in multiples of 15 rows at each timepoint), was 4312322133314111311222332313213211 for the data shown in Fig. 4, and we provide more example sequences in the software package online (Code Availability). The sequence was designed so that any subsequence of 4 timepoints or longer was unique within the sequence to allow unambiguous matching. The final voltage pulse "1" at the end of the sequence triggered CCD readout, as the total count of triggers (68) reached the preset value on the camera. The EM-CCD Arm Output signal reported when the camera was ready to start the next VS-SpeedyTrack acquisition cycle and triggered the AWG to relaunch the shift sequence. The same vertical-shift pattern was thus repeatedly applied to each VS-SpeedyTrack frame. A second output channel of the AWG triggered a DAQ card (PCI-6733, National Instruments) to switch off the excitation (561 nm) laser during the readout phase of each frame to avoid unnecessary illumination and to switch on a weak activation (405 nm) laser for ~5 ms before each VS-SpeedyTrack frame to photoconvert a very small fraction of Dendra2 into the red state to be excited. The power of the 405 nm laser was gradually adjusted (~30–150 W/cm²) during the experiment to maintain a high enough count of single-molecule images across the recorded frame while avoiding trajectory overlapping. The data shown in Fig. 4 thus maintained ~120 single-molecule images/frame over 55,000 VS-SpeedyTrack frames acquired in ~80 min. An upper limit of ~300 single-molecule images per frame is recommended for VS-SpeedyTrack to allow correct decoding. Segmenting the acquired VS-SpeedyTrack dataset into sub-datasets of 10,000 frames showed stable ER shape and diffusion properties (Supplementary Fig. 9), demonstrating that diffusion mapping can be achieved with fewer frames.

## SpeedyTrack trajectory reconstruction and analysis

For each SpeedyTrack frame, single-molecule images were first identified and localized using GDSC SMLM[50]. The resultant localizations were then grouped into trails using custom MATLAB scripts (Code Availability). Starting with a localization at the bottom of the frame, the algorithm searched for the next timepoint within a preset radius $r$ (~800 nm typical) of the equivalent position defined by deducting from the previous localization the experimentally executed vertical shift $\delta$ between timepoints (typically 15 rows). The search radius $r$ was set to[51] $2.55\sqrt{4D'\Delta t}$, where $D'$ is the largest expected diffusion coefficient for the sample. For the found localization, the code continued to

search for the next timepoint after deducting another $\delta$ from the new localization, and the process repeated. The $\delta$-subtracted identified locations were then concatenated into a single-molecule trajectory. In any timestep, if no localization was found within the search radius, the code introduced a "skipped" data point and flagged as such so that these placeholders allowed the continued searching for the next timepoint in the streak but were not included in the actual trajectory for SPT analysis. This treatment helped handle missed localizations, *e.g.*, due to blinking, temporary out of focus, and VS-SpeedyTrack gaps (below). Trajectories were terminated when multiple "skipped" data points were found in a row (3 or more for typical SpeedyTrack with fixed vertical shifting and 6 or more for VS-SpeedyTrack), and the skips at the end were removed. Trajectories that were less than 4 single-molecule observations or dominated by "skipped" data points were discarded. After the termination of a trajectory, localizations used in the trajectory were removed from the pool of localizations, and the search for a new trajectory began using the next available localization at the bottom of the frame. This process was repeated until all localizations in a frame were assigned to a trajectory or discarded.

The reconstructed trajectories were analyzed following typical SPT treatments. For in vitro experiments, MSD values at varying time lags $t_{lag}$ were calculated from pooled trajectories. $D$ was determined by a linear fit of the first four timepoints to MSD = $4Dt_{lag} + b$, wherein the intercept $b$ accounts for single-molecule localization uncertainties (static error) and motion blur (dynamic error), which respectively contribute positively and negatively to the $b$ value[26,27]. For FRET analysis, single-molecule images in the donor and acceptor channels were separately localized and then mapped to the same coordinate system before trajectories were reconstructed as described above. FRET efficiency $E_{FRET}$ was then calculated for each timepoint as the detected single-molecule intensity in the acceptor channel divided by the sum intensity of both channels. DNA hairpin opening-closing kinetics were analyzed by taking observations with $E_{FRET} > 0.65$ and $E_{FRET} < 0.45$ as closed and open states, respectively. Observations with $0.45 < E_{FRET} < 0.65$ or "skipped" localizations were excluded from the analysis. Time-correlated conditional probability was calculated from all FRET time traces as $P_{cc}(t_{lag}) = P(\text{closed}, t+t_{lag} \mid \text{closed}, t)$: Namely, for all closed-state observations, we counted the number of times the same hairpin was open or closed at $t_{lag}$ later. The use of conditional probabilities, rather than the dwell times in each state, avoids complications from short and donor-only labeled trajectories. Hairpin opening and closing rates, $k_{open}$ and $k_{close}$, were determined by fitting the correlation results to an exponential decay model $P_{cc} = (1-F_c) \exp(-k_{eff}t_{lag}) + F_c$, in which the two fitting parameters, the effective decay rate $k_{eff}$ and the fraction of closed state at equilibrium $F_c$, are related to $k_{open}$ and $k_{close}$ by $k_{eff} = k_{open} + k_{close}$ and $F_c = k_{close}/k_{eff}$. To generate two-dimensional distributions of FRET efficiency-diffusivity for the DNA strand hybridization data, single-molecule trajectories longer than 12 timepoints were each calculated for a mean FRET value and estimated for a diffusion coefficient using saSPT[36].

## VS-SpeedyTrack recovery of absolute positions and super-resolution mapping of trajectories

For the VS-SpeedyTrack data, single-molecule trajectories were first reconstructed as above. For every trajectory that contained 5 or more detected single-molecule images, the observed pattern of detection (1) vs. gap (0) was compared to the expected master sequence (Fig. 4b and Supplementary Fig. 7). Here, as each exposure gave a detected single-molecule image ("1") while the $m$ shifts after the exposure gave $m-1$ gaps ("0 s"), the $m = 4, 3, 2$, and 1 shifts in the VS-SpeedyTrack shifting scheme translated into barcodes of 1000, 100, 10, and 1, respectively, in the readout data. The full sequence above (4312322133314111311222332313213211) thus translated to a master barcode sequence of 1000100110100101011001001001100011110011101010010010010010011011. A Hamming distance was calculated

as the number of mismatches for each possible starting timepoint of the observed sequence in the master sequence. A successful alignment was attained when a starting timepoint uniquely achieved a smallest Hamming distance that was < 30% of the length of the observed sequence, which was achieved for ~85% of the trajectories. For each aligned trajectory, absolute $y$-positions were recovered by deducting the matched patterned vertical shifts in the sequence, thus generating a trajectory in real space. All single-molecule localizations from the resultant real-space trajectories were pooled together to generate an SMLM image. To map intra-ER diffusion, trajectories residing in the ER regions were calculated into single-molecule displacements at varied time lags of 1–10 timesteps. These time-varied displacements were spatially binned with a 120 nm grid for local statistics. For each spatial bin, a local principal diffusion direction was first determined based on single-molecule displacements at 2 ms (4 timesteps) time separation by averaging the bidirectional diffusion directions of all displacements in the bin[45]. The time-varied displacements in the bin were then calculated into MSD values and decomposed into two components parallel and perpendicular to the local principal direction, $MSD_{\parallel}$ and $MSD_{\perp}$. A colormap of local $D$ was then generated by fitting the first 4 data points in the resultant $MSD_{\parallel}$-$t_{lag}$ relationship in each bin to a one-dimensional normal diffusion model $MSD_{\parallel} = 2Dt_{lag} + b$. For examination of local diffusion modes, for each spatial bin, $MSD_{\parallel}$ values of the first 6 time lags were fit to the power law $MSD_{\parallel} = 2Kt_{lag}^{\alpha} + b$, for which we used the two initial data points to first determine the intercept $b$ based on a linear fit, and then performed a linear fit of $\ln(MSD_{\parallel} - b)$ against $\ln t_{lag}$ to determine $\alpha$ as the slope.

### Reporting summary

Further information on research design is available in the Nature Portfolio Reporting Summary linked to this article.

## Data availability

All data supporting the conclusions of this study are provided in the main figures and Supplementary Information. Raw data, as single-molecule localizations in the acquired frames, are made available at: doi.org/10.5281/zenodo.17187696.

## Code availability

Codes used in this work are made available at: doi.org/10.5281/zenodo.17187696.

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

## Acknowledgements

We acknowledge support by the National Institute of General Medical Sciences of the National Institutes of Health (R35GM149349, K.X.), the National Science Foundation (CHE-2203518, K. X.), the Arnold & Mabel Beckman Foundation through the Arnold O. Beckman Postdoctoral Fellowship in Chemical Instrumentation (M.A.S.), the Packard Fellowships for Science and Engineering (K.X.), and the Heising-Simons Faculty Fellows Award (K.X.).

## Author contributions

M.A.S. developed SpeedyTrack software and performed experiments. M.A.S. and K.X. designed experiments, analyzed results, and wrote the manuscript. K.X. supervised the project.

## Competing interests

M.A.S. and K.X. are inventors on a US provisional patent application No. 63/783,718 related to SpeedyTrack.
