## [Peer Review File · Nature Communications]

Direct microsecond wide-field single-molecule tracking and super-resolution mapping via CCD vertical shift

Corresponding Author: Professor Ke Xu

Version 0:

Reviewer comments:

Reviewer #1

(Remarks to the Author)

The present work is presented by an expert group in single molecule imaging, particularly also PALM/STORM and single molecule tracking approaches. They have in the past developed technical tricks to further improve on temporal resolution and the detection of high densities of highly diffusive particles and the generation of statistically relevant data on the diffusivity in cellular compartments. In the present work, the authors make use of the intrinsic mechanism of EMCCD cameras to transfer data from the light-exposed area of the camera chip to the readout area, which occurs much faster than the pixel-by-pixel readout of the data from the camera. By computationally manipulating the camera through dedicated software, they transfer a small readout "slice" of the chip area after tens of microseconds of exposure to the readout area. They do so repeatedly after subsequent exposures such as that the same area is stored always shifted downward by i.e. 10 rows many times on the chip until it is read-out. In this way, if the molecules move less than on average 5 pixels per frame, they be found in the next image 10 rows below the first exposure, thus allowing for single molecule tracking from the "stacked" exposures in the field of view. Doing SPT this way, they achieve measurements of diffusivities of well into 1000 $\mu\text{m}^2/\text{s}$. They then use test their method using a number of nice examples and extend it to single molecule FRET at the example of DNA-oligomer interactions. They yield thousands of measurements and can demonstrate salt-dependent zippering of an oligomer with a dye-dye FRET pair. They also show that if one of two dye coupled oligomers is much larger than the other, FRET is detected only for the much heavier dimer. Of course the problem with this is that as the transfer frame of the camera is overwritten time and again, molecules may appear newly at any time and location in the image and thus be newly transferred in the next overwriting transfer. As a result, it may not be clear if this detected molecule stems from the location it is detected at or from whatever n-th shift. To overcome this problem, the authors have devised a clever trick: they vary the amount of lines the next integration is shifted onto the transfer frame in a particular sequence (i.e. 4x, 2x, 2x, 3x, 1x, 1x,) such that a molecule from its first appearance must fit in its subsequent localizations somehow into that sequence to allow the correct determination of its position. In this way, the authors could use the technique to take images of microsecond diffusion of solute FPs in the ER.

This reviewer is convinced that the presented work is of great importance for the analysis of protein and solute mobility in cells and provides unprecedented access to the analysis of single molecule mobility on cells at microsecond resolution. It should be published.

This is excellent work that is very well documented and presented. The reviewer appreciates the presentation of raw data in the supplement and the detailed explanation of the procedure.

(Remarks on code availability)

Reviewer #2

(Remarks to the Author)

This manuscript describes the development of a method, termed SpeedyTrack, for wide-field single-molecule tracking super-resolution mapping. The developed method has achieved with a maximum temporal resolution of 50 μs and up to probably 58 steps or 40 steps (timepoints; see below). The authors employ a method to leverage the vertical shift function of

commercially available EM-CCD cameras. By projecting the time domain into the spatial domain, very fast single-molecule tracking became possible. By deconvoluting spatial information with a temporally patterned vertical shift “barcode,” the technique facilitates single-molecule localization microscopy (SMLM).

While the basic idea of using the vertical shift feature of CCD cameras for fast single-molecule imaging is not entirely new (as demonstrated by the Schütz group, Ref. 18, which reported 50- μ s exposures, shifts taking up to 450 μ s, providing a temporal resolution of 500 μ s, 10-fold slower than that described here, due to earlier technological limitations 18 years ago), its applications to single-molecule tracking with sufficient temporal resolution to detect free diffusion of fluorophores in 3D liquids as well as to SMLM, has not been previously accomplished.

The significance of SpeedyTrack lies in its ability to achieve simultaneously track multiple single molecules at high speed using only the standard functions of commercially available EM-CCD cameras, without requiring modifications to the camera or optical system (compared to simultaneous tracking of only one or a few molecules in the case of MINFLUX). This capability could render the technique extremely useful for applications in which tracking limitations of several tens of steps are acceptable.

SpeedyTrack was also found useful to observe smFRET, allowing simultaneous observations of single-molecule diffusion and FRET states.

This manuscript also describes a very smart invention for its application to obtaining SMLM images. It introduces a temporally patterned vertical shifting scheme to encode time, enabling the recovery of temporal information and the deconvolution to obtain absolute positions of observed fluorescent spots (VS-SpeedyTrack).

If the high time-resolutions with single-molecule sensitivities claimed in the manuscript could be achieved with widely available cameras, this method could significantly broaden access to fast single-molecule imaging and mapping, making it a valuable tool for wide ranges of scientists. However, we co-reviewers believe that several fundamental issues regarding the method’s applicability to single-molecule imaging and SMLM require further clarification.

(1) General concerns, mainly related to Figures 2 and 4.

1a) Lack of the method validation for proper detection of single molecules

Almost all the tests for validating the developed method are based on the average diffusion behaviors obtained from many molecules, which are shown in Figure 2 (please include the number of observed trajectories in Figures 2f, 2i, and 2j, along with error bars for individual data points. If the error bars are smaller than the symbols (perhaps due to the measurements of many molecules), please state this explicitly; The diffusion coefficients also require the standard error of the mean (SEM); please indicate the temperature used for the experiments; for only the panel of the acetone-PBS data, “room temperature” is mentioned, but an exact temperature should be provided because the solvent viscosity could be very sensitive to temperature).

However, the most interesting applications of this method would involve examining the behaviors of individual molecules, which might reveal complex regulation mechanisms for the movements of the observed molecules. Therefore, it is essential to assess the method’s performance at the single-molecule level, i.e., we will need to know how well this method can perform in the fast observations of single molecules (short integration times).

One of the key issues with very fast single fluorescent-molecule imaging and tracking is the number of photons that can be obtained from a single dye molecule during a single frame time, like 50 μ s in the case of present manuscript. In their typical experiments, they should clarify the distribution of photon numbers from a single dye molecule for a single integration time period. Please include 50- μ s data for this.

At a minimum, please provide single-molecule localization precisions at several observation frame rates (or integration times), particularly in the integration time ranges of 50-800 μ s. Without this information, it is impossible to fully assess the utility of this method for single-molecule studies. Simply obtaining the diffusion coefficient averaged over many single molecules in the bulk medium would not be the most interesting and exciting applications of single-molecule imaging experiments.

1b) Difficulty for validating single-molecule imaging capabilities using 3D solutions and pseudoTIRF or HILO illuminations

The videos in Supplementary Information suggest considerable spatial variations in laser intensity across the illuminated area, possibly due to the use of pseudo-TIRF/HILO illumination. This non-uniformity will make validations for single-molecule imaging and SMLM quite difficult, as we requested in our point 1a). To clarify the point 1a) and the following points, we would recommend measurements using immobilized specimens under more uniform TIRF illumination to determine the photon counts and localization precisions. This would not be difficult for the authors. To address these points, please consider using immobilized specimens. Alternatively, if these measurements are performed in 3D liquids, the authors should address the effects of non-uniform illumination. The authors emphasize wide-field microscopy, but we guess that they do not try to mean only the measurements in 3D liquids.

1c) Another way of validating SpeedyTrack for single-molecule imaging and SMLM

Related to 1a), one of the ways to validate their method would be to plot the single-molecule localization precisions as a function of the number of detected photons from single molecules for a single timepoint (they will need to vary the excitation laser intensity). Fitting the data using Mortensen’s equation to verify that the F value is close to $\sqrt{2}$ (the expected value for EMCCD) would tremendously strengthen the authors’ argument that SpeedyTrack can enhance the time resolution to 50 μ s.

1d) Evaluation for three fluorescent molecules

It would be excellent to address the points 1a)-1c) for Cy3B, Atto647N, and Dendra2 with information of the employed laser power densities. If other dyes were tested before ultimately deciding on these fluorescent probes, information about the dyes tested would also be valuable.

1e) Accumulation of background noise

Since the same frame is exposed multiple times (up to 68 times), the background noise accumulates. The authors should provide an analysis of the signal-to-noise ratio to illustrate how background levels increase with additional shifts and how this affects the single-molecule localization precision.

Meanwhile, in the explanation of the software usage, the authors explained that the top 1/7 of the imaging field is unusable due to noise. Therefore, under the standard conditions of 68 shifts, only 58 timepoints are usable. This should be noted in

the main text.

1f) Motion blur evaluation

The assumption that motion blur for freely diffusing molecules in 3D liquids is negligible at the short integration times used should be experimentally verified.

1g) Signal intensity measurements for individual fluorescent spots.

Related to the points 1a)-1e), we wonder whether SpeedyTrack can (or can be made to) correctly measure the intensities of individual spots. Such a capability would be required for proper single-molecule tracking and SMLM. These should be tested at a few integration times to tell their consistencies (please include 50- μ s data). Again, this test may need to be performed using immobilized specimens and uniform TIRF illuminations (or clarify the spatial variations of the illumination laser intensity used presently, and address this issue in some ways).

We consider that the following experiments could examine single-molecule imaging capabilities in a quantitative way. Since authors stated in the Methods section that in their labeling of carbonic anhydrase, the dye-to-protein ratio was 0.9 (we guess this is a mean value; please provide SEM). Following Poisson distribution, about 37% and 17% of the anhydrase molecules are tagged with one and two Cy3B probes, respectively. Therefore, a useful test would be to actually measure the intensity distribution and examine whether it could be decomposed into monomeric and dimeric distributions (+ small amounts of oligomeric intensities), and examine whether their populations represent approximately 2:1 ratio and also whether the monomeric intensity distribution matches with the intensity distribution of free Cy3B. Alternatively, the distribution for free Cy3B could be used to deconvolute the Cy3B-anhydrase intensity distribution, to examine whether it could be deconvoluted properly.

(2) Diffusion data shown in Figure 2e

Around 60-70% of the fluorescent spots they observed appear to disappear within 10 frames. Can the authors evaluate the fractions of the disappeared spots by the processes of photobleaching and diffusing out of the focal plane? Since photobleaching would be significant for very fast single molecule imaging due to the use of high laser power conditions, they should evaluate photobleaching rate under their observation conditions, perhaps by using immobilized molecules (see point 1b).

(3) smFRET data shown in Figure 3

3a) Sensitivity limitations

Related to our point 1a), we wonder the sensitivities of single molecules in FRET measurements might be lower.

The authors used an 800- μ s integration time or a frame rate of \approx 1.25 kHz. Is this because of lower sensitivities of smFRET measurements? It is necessary to clarify whether this slower frame rate reflects a limitation in sensitivity for smFRET.

This slower rate brings up further questions. Can the authors still track single hairpin DNA molecules at this slow rate, or did they only observe those diffusing slowly at the time of observations? Is the diffusion coefficient of these molecules measured at this rate the same as that observed with an integration time of 50 or 300 μ s? If the diffusion coefficient measured here is smaller than that for carbonic anhydrase, please explain why. Any data about Stokes' radius for these molecules?

The Abstract and Introduction currently give the impression that smFRET can be observed with a 50- μ s integration time. Is this possible? If it is not feasible, the text should be revised for clarity.

3b) Data analysis in Figures 3c and 3e

In the derivation of Pcc shown in Figure 3e, the data included all trajectories that lasted 12 timepoints or longer. This probably means that some trajectories always exhibited the closed state and some trajectories exhibited multiple transitions between closed and open states. We suspect that the Pcc dependence on the time lag (the data shown in Figure 3e) might be skewed by the limited observation period (i.e., the decay time constants might be shortened due to the inclusion of shorter trajectories). This point should be clarified. In addition, histograms showing the closed and open state durations (including the number of trajectories that did not exhibit any state transitions) should be shown in the figure. This will help clarify the point raised here. In addition, by measuring the trajectory duration histograms, this problem could be theoretically resolved. This should also be tried.

(4) Figures 1d and 4a-c

4a) Please provide the estimate of approximate upper limit of the molecular densities when performing SMLM by VS-SpeedyTrack.

4b) According to the descriptions in Methods and Supplementary Video 2, more than 2,200 seconds (55,000 frames x 40 ms; \approx 37 min) might have been spent to obtain this image. This duration appears too lengthy for observing the rapidly changing shape of the ER, which might occur in time scales of seconds. We suspect that the clear nice image displayed in Fig. 4d might have been obtained by shorter data acquisition time. Please clarify.

Minor points

1. Regarding the software, we have confirmed that the SpeedyTrack Acquisition GUI operates without any issues. As for tracking and analysis, it is difficult to test without actual single-molecule vertical shift images, so it would be better to also provide real vertical shift images like those used in the Supplementary Videos (i.e., demo data) on the software distribution site.

2. Color usage in figures. Please use color combinations by which color blind readers can readily understand the figures.

(Remarks on code availability)

Reviewer #3

(Remarks to the Author)

SpeedyTrack is an interesting idea to locally encode the time dimension into the spatial dimension of a CCD camera. Encoding time into the spatial dimension is not entirely new (see e.g. doi.org/10.1021/ac100302s), but to my knowledge it

has not been done for diffusion analysis. The idea behind the method is that charges can be shifted extremely fast on the camera, while the readout is comparatively slow. It is similar to the well-known kinetic mode in CCD camera operation, but what is new here is that the authors propose to shift the images by extremely small distances, so that the images partially overlap. As the authors correctly point out, this only works at low molecular concentrations. The advantage of SpeedTrack is that i) such small shifts can be performed at very high speed, and ii) long trajectories can be recorded in one run before the camera chip is slowly read out. The disadvantage is that molecular densities must be very low to avoid signal overlap. Encoding time in the spatial dimension is not entirely new (see e.g. doi.org/10.1021/ac100302s), but diffusion analysis has not, to my knowledge, been done before. While the idea is intriguing, there are a number of issues that I believe limit the practical use of this method, or at least should be discussed.

1.) The short shift time of $\sim 5\mu\text{s}$ (for 10 lines) is only advantageous if it is comparable to the illumination time (otherwise the time resolution is more or less limited by the illumination). Most researchers use illumination times of tens of milliseconds; the use of millisecond illumination is rare, sub-millisecond illumination (as used here) is hardly used. Reducing the illumination time by a factor of x also reduces the signal-to-noise ratio. This can be compensated by increasing the laser intensity by the same factor x , but only as long as the excitation intensity remains well below the saturation intensity of the fluorophore. An exposure time of $50\mu\text{s}$, as reported here, seems to be too short to be realistic. What is missing in this manuscript is: i) a rationale as to why $50\mu\text{s}$ illumination provides sufficient signal to detect single dye molecules; ii) a clear statement that the advantage of this method lies in such exceptionally short illumination times.

2.) The fluorophore density limitation should be discussed quantitatively. The method requires time traces to remain clean for the duration of the experiment, without overlap with other molecules.

3.) In many applications in this paper, the illumination was actually not that short. For example, in Figures 2a-f and 3 the authors used $300\mu\text{s}$. I think a shift time of $25\mu\text{s}$ would still be OK for such a setting, which would allow standard kinetics mode readout of images of ~ 50 lines. This would still give 20 images per sequence (instead of 100 images for the SpeedyTrack), which may still be ok considering limit track length due to photobleaching. The advantage of the standard kinetics mode would be that the density of molecules can be much higher (see my point 2).

4.) A problem occurs when molecules blink, as there is an ambiguity between time and y -position. This effect is particularly problematic in single molecule localization microscopy. The authors propose a decoding scheme by introducing additional different shifts between two recorded images. I think that this decoding is only unambiguous if the shift intervals vary over the whole CCD chip; otherwise, how would one distinguish between blinks that are partially synchronized with the applied barcode (which can happen from time to time)? Moreover, decoding obviously requires additional shifts, which introduces long gaps in the image sequence, further reducing the advantage of the method over standard kinetic mode.

(Remarks on code availability)

Reviewer #4

(Remarks to the Author)

(Remarks on code availability)

Version 1:

Reviewer comments:

Reviewer #2

(Remarks to the Author)

The authors have responded satisfactorily to most of the points we raised in the first round. However, several critical issues remain inadequately addressed. These are detailed below.

1e) Accumulation of background noise

We consider that the added sentence in the main text is insufficient. The authors should discuss how the accumulation of background noise may impact the quality and interpretation of single molecule tracking data.

1g) Signal intensity measurements for individual fluorescent spots

The authors misunderstood some of the points we raised. Given that the dye-to-protein ratio determined by absorption spectroscopy was 0.9, their Cy3B-labeled carbonic anhydrase specimen is considered to include the following populations (as calculated using a Poisson distribution with a mean of 0.9): 37% of the protein molecules are tagged with one Cy3B molecule, 17% with two Cy3B molecules, and approximately 46% are unlabeled. This implies that around one-third of the observed fluorescent spots ($17/[17+37]$) represent protein tagged with two Cy3B molecules. This is a substantial fraction and

raises concerns regarding the validity of referring to this as single-molecule tracking to call it single fluorescent-molecule tracking.

Since this experiment is quite straightforward, we recommend that the authors repeat the experiments using a dye-to-protein ratio below 0.1 (only $\approx 5\%$ of the fluorescent spots represent 2 or more Cy3B molecules).

Note that even at a mean dye-to-protein ratio of 0.4, which is a value used by the authors for Cy3B-avidin (New Supplementary Fig. 3), 17% of the fluorescent avidin spots represented 2 or 3 Cy3B molecules. This could potentially explain the broad distributions shown in Supplementary Fig. 3a.

We therefore suggest that the authors repeat the experiments shown in Supplementary Fig. 3a using a mean dye-to-protein ratio below 0.1 and compare the distribution data with the present ones obtained for the dye-to-protein ratio of 0.4. This would allow the authors to demonstrate whether it is possible to resolve the fractional contributions of monomers and dimers coexisting in the specimen, and more specifically to identify the $\approx 17\%$ fluorescent spots arising from 2 or more Cy3B molecules.

In Supplementary Fig. 3, please also indicate error bars and the number of observations.

(2) Diffusion data shown in Figure 2e

To demonstrate that SpeedyTrack is broadly applicable to single molecule imaging and tracking, the authors should present the photobleaching kinetics data. This could readily be obtained by using immobilized Cy3B-avidin labeled at dye-to-protein ratios less than 0.1.

3b) Data analysis in Figures 3c and 3e

The analysis method based on Ref 32 (Kim et al., PNAS 2002) is, in our view, only valid under the conditions where photobleaching is negligible. We suspect that the authors' view is that photobleaching lifetime would be negligible compared with the dwell lifetime of the molecules in the observation volume. However, under the conditions of strong excitation laser illumination, photobleaching might not be negligible. This point has to be explicitly clarified, by measuring the photobleaching lifetimes particularly for observations at higher time resolutions.

(4) Figures 1d and 4a-c

4a) Estimation of the upper limit of the molecular densities for SMLM by VS-SpeedyTrack

As also recommended by Reviewer 3, we would feel much more comfortable if the authors could provide more quantitative estimates.

4b) To clarify the time courses for general readers, please also indicate the actual time in addition to the number of frames.

(Remarks on code availability)

The code appears to work well.

Reviewer #3

(Remarks to the Author)

The authors addressed all my points adequately. I recommend publication.

(Remarks on code availability)

Reviewer #4

(Remarks to the Author)

(Remarks on code availability)

Reviewer #1 (Remarks to the Author):

The present work is presented by an expert group in single molecule imaging, particularly also PALM/STORM and single molecule tracking approaches. They have in the past developed technical tricks to further improve on temporal resolution and the detection of high densities of highly diffusive particles and the generation of statistically relevant data on the diffusivity in cellular compartments. In the present work, the authors make use of the intrinsic mechanism of EMCCD cameras to transfer data from the light-exposed area of the camera chip to the readout area, which occurs much faster than the pixel-by-pixel readout of the data from the camera. By computationally manipulating the camera through dedicated software, they transfer a small readout "slice" of the chip area after tens of microseconds of exposure to the readout area. They do so repeatedly after subsequent exposures such as that the same area is stored always shifted downward by i.e. 10 rows many times on the chip until it is read-out. In this way, if the molecules move less than on average 5 pixels per frame, they be found in the next image 10 rows below the first exposure, thus allowing for single molecule tracking from the "stacked" exposures in the field of view. Doing SPT this way, they achieve measurements of diffusivities of well into $1000 \mu\text{m}^2/\text{s}$. They then use test their method using a number of nice examples and extend it to single molecule FRET at the example of DNA-oligomer interactions. They yield thousands of measurements and can demonstrate salt-dependent zippering of an oligomer with a dye-dye FRET pair. They also show that if one of two dye coupled oligomers is much larger than the other, FRET is detected only for the much heavier dimer. Of course the problem with this is that as the transfer frame of the camera is overwritten time and again, molecules may appear newly at any time and location in the image and thus be newly transferred in the next overwriting transfer. As a result, it may not be clear if this detected molecule stems from the location it is detected at or from whatever n-th shift. To overcome this problem, the authors have devised a clever trick: they vary the amount of lines the next integration is shifted onto the transfer frame in a particular sequence (i.e. 4x, 2x, 2x, 3x, 1x, 1x,) such that a molecule from its first appearance must fit in its subsequent localizations somehow into that sequence to allow the correct determination of its position. In this way, the authors could us the technique to take images of microsecond diffusion of solute FPs in the ER.

This reviewer is convinced that the presented work is of great importance for the analysis of protein and solute mobility in cells and provides unprecedented access to the analysis of single molecule mobility on cells at microsecond resolution. It should be published.

This is excellent work that is very well documented and presented. The reviewer appreciates the presentation of raw data in the supplement and the detailed explanation of the procedure

Response: We thank the reviewer for their excellent summary and enthusiasm for our work.

Reviewer #2 (Remarks to the Author):

This manuscript describes the development of a method, termed SpeedyTrack, for wide-field single-molecule tracking super-resolution mapping. The developed method has achieved with a maximum temporal resolution of $50 \mu\text{s}$ and up to probably 58 steps or 40 steps (timepoints; see below). The authors employ a method to leverage the vertical shift function of commercially available EM-CCD cameras. By projecting the time domain into the spatial domain, very fast single-molecule tracking became possible. By deconvoluting spatial information with a temporally patterned vertical shift "barcode," the technique facilitates single-molecule localization microscopy (SMLM).

While the basic idea of using the vertical shift feature of CCD cameras for fast single-molecule imaging is not entirely new (as demonstrated by the Schütz group, Ref. 18, which reported $50\text{-}\mu\text{s}$ exposures, shifts taking up to $450 \mu\text{s}$, providing a temporal resolution of $500 \mu\text{s}$, 10-fold slower than that described here,

due to earlier technological limitations 18 years ago), its applications to single-molecule tracking with sufficient temporal resolution to detect free diffusion of fluorophores in 3D liquids as well as to SMLM, has not been previously accomplished.

The significance of SpeedyTrack lies in its ability to achieve simultaneously track multiple single molecules at high speed using only the standard functions of commercially available EM-CCD cameras, without requiring modifications to the camera or optical system (compared to simultaneous tracking of only one or a few molecules in the case of MINFLUX). This capability could render the technique extremely useful for applications in which tracking limitations of several tens of steps are acceptable.

SpeedyTrack was also found useful to observe smFRET, allowing simultaneous observations of single-molecule diffusion and FRET states.

This manuscript also describes a very smart invention for its application to obtaining SMLM images. It introduces a temporally patterned vertical shifting scheme to encode time, enabling the recovery of temporal information and the deconvolution to obtain absolute positions of observed fluorescent spots (VS-SpeedyTrack).

If the high time-resolutions with single-molecule sensitivities claimed in the manuscript could be achieved with widely available cameras, this method could significantly broaden access to fast single-molecule imaging and mapping, making it a valuable tool for wide ranges of scientists. However, we co-reviewers believe that several fundamental issues regarding the method's applicability to single-molecule imaging and SMLM require further clarification.

Response: We thank the co-reviewers for their kind summary and assessment of the manuscript. We provide a point-by-point response to their comments below. For the point of “widely available cameras”, we note that most of the data we present are based on Andor iXon Ultra 897, which we believe is one of the most popular cameras in single-molecule microscopy. We also showed that our approach is readily implemented with another platform (Nuvu).

(1) General concerns, mainly related to Figures 2 and 4.

1a) Lack of the method validation for proper detection of single molecules

Almost all the tests for validating the developed method are based on the average diffusion behaviors obtained from many molecules, which are shown in Figure 2 (please include the number of observed trajectories in Figures 2f, 2i, and 2j, along with error bars for individual data points. If the error bars are smaller than the symbols (perhaps due to the measurements of many molecules), please state this explicitly; The diffusion coefficients also require the standard error of the mean (SEM); please indicate the temperature used for the experiments; for only the panel of the acetone-PBS data, “room temperature” is mentioned, but an exact temperature should be provided because the solvent viscosity could be very sensitive to temperature).

Response: We have updated the caption of **Figure 2** to include the number of observed trajectories and standard errors. **Figure 2f, i** have been updated to show error bars based on the MSD at each time lag calculated for each trajectory (and weighted by trajectory length). Errors were calculated for each independent run of 1,000-15,000 tracks shown in **Figure 2j**; error bars are displayed on the figure for points where they are larger than the symbol size. We also have added a note to the Methods section to indicate the temperature at which measurements were performed ($23\pm 0.5^\circ\text{C}$).

However, the most interesting applications of this method would involve examining the behaviors of individual molecules, which might reveal complex regulation mechanisms for the movements of the

observed molecules. Therefore, it is essential to assess the method's performance at the single-molecule level, i.e., we will need to know how well this method can perform in the fast observations of single molecules (short integration times).

One of the key issues with very fast single fluorescent-molecule imaging and tracking is the number of photons that can be obtained from a single dye molecule during a single frame time, like 50 μ s in the case of present manuscript. In their typical experiments, they should clarify the distribution of photon numbers from a single dye molecule for a single integration time period. Please include 50- μ s data for this.

At a minimum, please provide single-molecule localization precisions at several observation frame rates (or integration times), particularly in the integration time ranges of 50-800 μ s. Without this information, it is impossible to fully assess the utility of this method for single-molecule studies. Simply obtaining the diffusion coefficient averaged over many single molecules in the bulk medium would not be the most interesting and exciting applications of single-molecule imaging experiments.

Response: Thank you for this discussion. Our focus is on improving the detector-limited temporal resolution of single-molecule imaging. SpeedyTrack offers such an improvement over traditional methods that the temporal resolution is now limited by probe brightness. We note that using brighter probes (e.g., nanoparticles), time resolutions even less than 50 μ s may be possible.

To assess the photon count and localization precisions for the Cy3B dye we mostly used in this work, we immobilized Cy3B-labeled avidin on the coverslip and performed SpeedyTrack at various exposure times (New **Supplementary Fig. 3**). At 50 μ s exposure times, the photon counts per exposure for each molecule exhibited a distribution with a mean of 145 photons. Increased photon counts were observed for longer exposures. Using SpeedyTrack, the same immobilized molecule was measured multiple times; scattering of the x and y positions represented uncertainty in the localization. We repeated this process for many molecules at various exposure times to quantify how the single-molecule localization precision depended on the integration time. These results confirm our ability to successfully localize single molecules even at short exposure times. We have added related discussion to Page 6: "While the increased temporal resolution of SpeedyTrack limits the detected photons per localization, for Cy3B we were able to collect >100 photons in 50 μ s to achieve a localization precision of ~30 nm (**Supplementary Fig. 3**). Longer exposure times and/or higher excitation powers yielded higher photon counts and precision ... Use of brighter fluorophores (e.g., nanoparticles) should allow higher spatial resolution."

1b) Difficulty for validating single-molecule imaging capabilities using 3D solutions and pseudoTIRF or HILO illuminations

The videos in Supplementary Information suggest considerable spatial variations in laser intensity across the illuminated area, possibly due to the use of pseudo-TIRF/HILO illumination. This non-uniformity will make validations for single-molecule imaging and SMLM quite difficult, as we requested in our point 1a). To clarify the point 1a) and the following points, we would recommend measurements using immobilized specimens under more uniform TIRF illumination to determine the photon counts and localization precisions. This would not be difficult for the authors. To address these points, please consider using immobilized specimens. Alternatively, if these measurements are performed in 3D liquids, the authors should address the effects of non-uniform illumination. The authors emphasize wide-field microscopy, but we guess that they do not try to mean only the measurements in 3D liquids.

Response: As suggested by the reviewers, we focus on TIRF illumination and molecules immobilized on a coverslip as we addressed their points in 1a–c. These results are shown in **Supplementary Fig. 3**.

1c) Another way of validating SpeedyTrack for single-molecule imaging and SMLM

Related to 1a), one of the ways to validate their method would be to plot the single-molecule localization precisions as a function of the number of detected photons from single molecules for a single timepoint (they will need to vary the excitation laser intensity). Fitting the data using Mortensen's equation to verify that the F value is close to $\sqrt{2}$ (the expected value for EMCCD) would tremendously strengthen the authors' argument that SpeedyTrack can enhance the time resolution to 50 μ s.

Response: We have performed experiments with a fixed exposure time of 0.3 ms in which we varied the laser power. Because the background level and the photon counts are both sensitive to illumination intensity, we plot the measured localization precision versus that predicted by Mortensen's equation (σ_0) calculated using the average photons per localization and background at different excitation powers (**Supplementary Fig. 3**). We find that the measured values ($\sigma_{measured}$) indeed followed the expected trend of $\sigma_{measured} = \sqrt{2} \times \sigma_0$, with the EMCCD multiplicative noise giving a scaling factor of $\sqrt{2}$.

We have added related discussion to Page 6: "Longer exposure times and/or higher excitation powers yielded higher photon counts and precision, and the measured localization errors at different photon counts and background levels agreed with theoretical predications³¹ after accounting for the EMCCD multiplicative noise (**Supplementary Fig. 3**)."

1d) Evaluation for three fluorescent molecules

It would be excellent to address the points 1a)-1c) for Cy3B, Atto647N, and Dendra2 with information of the employed laser power densities. If other dyes were tested before ultimately deciding on these fluorescent probes, information about the dyes tested would also be valuable.

Response: In our response to points 1a-c above, we focused on Cy3B as this is the fluorophore used in the majority of our experiments. Cy3B was chosen for its high brightness and easy labeling of most protein targets. Cy3B is also one of the most used dyes for single-molecule experiments. We have included information about the employed power densities in **Supplementary Fig. 3**. In our experiments, Atto647N is only used as an acceptor in FRET experiments, and therefore has a more complex signal than for direct excitation. We have added further FRET results at faster time resolutions in this revision (discussed below), which we believe offer additional evidence of the ability of the method. For Dendra2, we do not have a good way to immobilize single molecules. However, since the emission spectrum of Dendra2 is similar to Cy3B, we estimated a localization precision of ~ 25 nm in our data in Figure 4 based on the detected single-molecule photon count (150 photons per localization) and background (~ 3.5 photons/pixel). We have added this information to the text.

1e) Accumulation of background noise

Since the same frame is exposed multiple times (up to 68 times), the background noise accumulates. The authors should provide an analysis of the signal-to-noise ratio to illustrate how background levels increase with additional shifts and how this affects the single-molecule localization precision.

Meanwhile, in the explanation of the software usage, the authors explained that the top 1/7 of the imaging field is unusable due to noise. Therefore, under the standard conditions of 68 shifts, only 58 timepoints are usable. This should be noted in the main text.

Response: To address the impact of background accumulation on the performance of SpeedyTrack, we focused on the metric of localization precision as discussed above. Cy3B-labeled protein was immobilized, and the localization precision was determined for a variety of shift heights. The exposure time was kept constant at 0.3 ms. As the background increases with smaller shift heights, the localization error also increases. We have added a discussion of this to page 6 and **Supplementary Fig. 4**: "Background accumulation during vertical shifts reduces the achievable localization precision; the use of

larger vertical shifts alleviates this issue at the expense of a decreased maximum track length (**Supplementary Fig. 4**).”

For the additional noise at the top portion of the frame, this is caused by continued exposure of the sensor during the readout period of the measurement. This can be eliminated by turning off the illumination during the readout period. We have clarified this point in the code instructions where this smearing effect is discussed.

1f) Motion blur evaluation

The assumption that motion blur for freely diffusing molecules in 3D liquids is negligible at the short integration times used should be experimentally verified.

Response: We clarify that we do not assume that motion blur is negligible in our data. Although not often discussed in the literature, motion blur (dynamic error) contributes a negative offset to the MSD-time lag curve whereas localization error adds a positive offset. See detailed discussion in Refs. 25 & 26. This contribution is accounted for in fitting the MSD curves to a line with an offset/intercept that accounts for both motion blur and localization error. Consequently, we were able to accurately determine D values for a wide range of samples and conditions, including the viscosity dependency for free Cy3B as shown in **Figure 2j**. In our new **Supplementary Fig. 5j**, we further show consistent D values obtained for SpeedyTrack experiments carried out at 0.3 and 0.8 ms time resolutions.

1g) Signal intensity measurements for individual fluorescent spots.

Related to the points 1a)-1e), we wonder whether SpeedyTrack can (or can be made to) correctly measure the intensities of individual spots. Such a capability would be required for proper single-molecule tracking and SMLM. These should be tested at a few integration times to tell their consistencies (please include 50- μ s data). Again, this test may need to be performed using immobilized specimens and uniform TIRF illuminations (or clarify the spatial variations of the illumination laser intensity used presently, and address this issue in some ways).

We consider that the following experiments could examine single-molecule imaging capabilities in a quantitative way. Since authors stated in the Methods section that in their labeling of carbonic anhydrase, the dye-to-protein ratio was 0.9 (we guess this is a mean value; please provide SEM). Following Poisson distribution, about 37% and 17% of the anhydrase molecules are tagged with one and two Cy3B probes, respectively. Therefore, a useful test would be to actually measure the intensity distribution and examine whether it could be decomposed into monomeric and dimeric distributions (+ small amounts of oligomeric intensities), and examine whether their populations represent approximately 2:1 ratio and also whether the monomeric intensity distribution matches with the intensity distribution of free Cy3B. Alternatively, the distribution for free Cy3B could be used to deconvolute the Cy3B-anhydrase intensity distribution, to examine whether it could be deconvoluted properly.

Response: We note that the absolute intensities of individual spots are not critical for single-molecule tracking or SMLM, and the distribution is often not ideally poissonian. For example, in STORM, we often observe an exponential decay for single-molecule brightness. The brightness is complicated by both blinking and illumination details. Multiple dyes labeled to the same protein further give complex inter-quenching behavior due to energy transfer. We further note that in this work, only the carbonic anhydrase sample may have multiple fluorophores. We clarify that the dye-to-protein ratio of 0.9 was for the specific sample we prepared for the measurement, measured using Nanodrop for bulk absorption spectroscopy. We have added this information to the Methods section. All other samples are intrinsically single-fluorophore as free dye, free FP, and DNA constructs with single-dye labeling, and SpeedyTrack worked well with tracking their trajectories. Further, our single-molecule FRET results demonstrate the ability to determine the relative intensities of two emission channels. Please see our new **Supplementary Fig. 3** for

the detected photon counts of immobilized Cy3B per localization as a function of integration time and excitation power, showing consistent increases for both longer integration times and higher illumination powers.

(2) Diffusion data shown in Figure 2e

Around 60-70% of the fluorescent spots they observed appear to disappear within 10 frames. Can the authors evaluate the fractions of the disappeared spots by the processes of photobleaching and diffusing out of the focal plane? Since photobleaching would be significant for very fast single molecule imaging due to the use of high laser power conditions, they should evaluate photobleaching rate under their observation conditions, perhaps by using immobilized molecules (see point 1b).

Response: We believe that for the case shown in Figure 2e, track length was limited by diffusion out of the focal plane. We performed a simple simulation to assess how long a molecule is expected to stay within the $\sim 0.8 \mu\text{m}$ focal depth of our microscope, assuming isotropic diffusion with $D = 90 \mu\text{m}^2/\text{s}$. The results are nearly identical to the experimentally measured distribution (new **Supplementary Fig. 2**). Of course, for other conditions photobleaching may become more dominant, but this will depend heavily on illumination conditions, dye characteristics, etc. We have added a discussion (Page 5): “The resultant trajectories, with typical durations of ~ 4 -40 timepoints following an exponential distribution likely limited by diffusion out of the focal plane (**Fig. 2e, Supplementary Fig. 2**),...”

(3) smFRET data shown in Figure 3

3a) Sensitivity limitations

Related to our point 1a), we wonder the sensitivities of single molecules in FRET measurements might be lower.

The authors used an 800- μs integration time or a frame rate of $\approx 1.25 \text{ kHz}$. Is this because of lower sensitivities of smFRET measurements? It is necessary to clarify whether this slower frame rate reflects a limitation in sensitivity for smFRET.

This slower rate brings up further questions. Can the authors still track single hairpin DNA molecules at this slow rate, or did they only observe those diffusing slowly at the time of observations? Is the diffusion coefficient of these molecules measured at this rate the same as that observed with an integration time of 50 or 300 μs ? If the diffusion coefficient measured here is smaller than that for carbonic anhydrase, please explain why. Any data about Stokes' radius for these molecules?

The Abstract and Introduction currently give the impression that smFRET can be observed with a 50- μs integration time. Is this possible? If it is not feasible, the text should be revised for clarity.

Response: For FRET experiments, we started with a longer exposure time for the hairpin sample in **Fig. 3a-f** considering that the FRET signal may be lower than direct excitation. However, the DNA hybridization results in **Figure 3h** were already obtained with a 300 μs exposure time.

In this revision, we show in **Supplementary Fig. 5** new results of DNA hairpin FRET data at 300 μs exposure time, which both clearly resolved switching between the open and closed conformations and yielded identical D values as measured at 800 μs exposure time. Both measurements result in a diffusivity of $\sim 58 \mu\text{m}^2/\text{s}$. This diffusion coefficient is reasonable for our 78-base DNA hairpin. Nkodo et al. report a diffusivity of $\sim 50 \mu\text{m}^2/\text{s}$ for a similarly sized DNA. In our new **Supplementary Fig. 6** we further demonstrate single-molecule FRET at an exposure time of 50 μs for DNA hybridization, which yielded FRET signal and D features comparable to the 300 μs data in **Figure 3h**. We have added these new results and discussions to the text.

3b) Data analysis in Figures 3c and 3e

In the derivation of Pcc shown in Figure 3e, the data included all trajectories that lasted 12 timepoints or longer. This probably means that some trajectories always exhibited the closed state and some trajectories exhibited multiple transitions between closed and open states. We suspect that the Pcc dependence on the time lag (the data shown in Figure 3e) might be skewed by the limited observation period (i.e., the decay time constants might be shortened due to the inclusion of shorter trajectories). This point should be clarified. In addition, histograms showing the closed and open state durations (including the number of trajectories that did not exhibit any state transitions) should be shown in the figure. This will help clarify the point raised here. In addition, by measuring the trajectory duration histograms, this problem could be theoretically resolved. This should also be tried.

Response: We clarify that our Pcc analysis was designed to avoid the potential biases mentioned by the reviewers, as well as other issues discussed below. We clarify that Pcc was calculated from all trajectories, not just those that lasted 12 timepoints or longer. The conditional probability includes normalization by the total number of useful paired observations at that time delay; therefore trajectories of all lengths can be used without skewing the results. This is also the reason we do not directly work with the observed closed and open state durations. For instance, if we have a relatively short trajectory of only 5 observations that remains closed for its entire duration, assigning a closed state duration of 5 underestimates the true closed state duration. Similarly, for the case of the molecule in **Fig. 3a,c**, which enters the FOV in a closed state, assigning a close state duration of 9 observations would again underestimate the closed-state duration because the hairpin was closed for an undetermined amount of time prior to entering the FOV. There is also the chance of a track having a gap where the state cannot be assigned, due to a missed localization or temporarily moving out of focus. These tracks would need to be discarded as it is possible the hairpin changed states during the gap. Finally, a fraction of the hairpin molecules lack the reporter dye and will artificially show as always being in the open state, further complicating the open-state duration analysis. For these reasons, we adopted the conditional probability Pcc analysis to evaluate, for each observed closed state of the hairpin, what the probability is for the same hairpin to be in the closed state at a later timepoint. We have added a statement in the Methods section to clarify this choice of analysis: “The use of conditional probabilities, rather than the dwell times in each state, avoids complications from short and donor-only labeled trajectories.”

(4) Figures 1d and 4a-c

4a) Please provide the estimate of approximate upper limit of the molecular densities when performing SMLM by VS-SpeedyTrack.

Response: We obtain ~120 localizations/frame for the VS-SpeedyTrack results in **Figure 4**. We recommend a density of <300 localizations per frame to avoid potential overlapping between nearby trajectories. We have added this discussion to the text and the Methods section.

4b) According to the descriptions in Methods and Supplementary Video 2, more than 2,200 seconds (55,000 frames x 40 ms; ~37 min) might have been spent to obtain this image. This duration appears too lengthy for observing the rapidly changing shape of the ER, which might occur in time scales of seconds. We suspect that the clear nice image displayed in Fig. 4d might have been obtained by shorter data acquisition time. Please clarify.

Response: Thank you for this discussion. This dataset is indeed ~55,000 frames. In our new **Supplementary Fig. 9**, we show that by segmenting this dataset into sub-datasets of 10,000 frames, stable ER shape and diffusion properties are observed over the full dataset. From this analysis, we further show that good D mapping can be achieved with much fewer frames. We have added this discussion to the Methods section.

Minor points

1. Regarding the software, we have confirmed that the SpeedyTrack Acquisition GUI operates without any issues. As for tracking and analysis, it is difficult to test without actual single-molecule vertical shift images, so it would be better to also provide real vertical shift images like those used in the Supplementary Videos (i.e., demo data) on the software distribution site.

Response: Thank you for this discussion. As the file sizes are large, in the submission it is difficult to provide data beyond what is already shown as examples (Supplementary Videos), but we will attempt to include example raw data online after publication.

2. Color usage in figures. Please use color combinations by which color blind readers can readily understand the figures.

Response: Thank you for pointing this out. We have revised **Figure 3** to use more accessible color schemes. We apologize that we have kept a rainbow color scheme in places where it is necessary to depict a large range of distinguishable values; however, we have tried to include other labels, arrows, etc. to improve the presentation.

Reviewer #3 (Remarks to the Author):

SpeedyTrack is an interesting idea to locally encode the time dimension into the spatial dimension of a CCD camera. Encoding time into the spatial dimension is not entirely new (see e.g. doi.org/10.1021/ac100302s), but to my knowledge it has not been done for diffusion analysis. The idea behind the method is that charges can be shifted extremely fast on the camera, while the readout is comparatively slow. It is similar to the well-known kinetic mode in CCD camera operation, but what is new here is that the authors propose to shift the images by extremely small distances, so that the images partially overlap. As the authors correctly point out, this only works at low molecular concentrations. The advantage of SpeedTrack is that i) such small shifts can be performed at very high speed, and ii) long trajectories can be recorded in one run before the camera chip is slowly read out. The disadvantage is that molecular densities must be very low to avoid signal overlap.

Encoding time in the spatial dimension is not entirely new (see e.g. doi.org/10.1021/ac100302s), but diffusion analysis has not, to my knowledge, been done before. While the idea is intriguing, there are a number of issues that I believe limit the practical use of this method, or at least should be discussed.

Response: We thank the reviewer for their kind comments on our work. As the reviewer points out, a few works have previously encoded time-domain information into the spatial dimension. However, they were based on the mechanical movement of the sample or the optics, the achieved time resolutions were orders of magnitude lower, and they have been largely limited to immobilized molecules. We have added a discussion to Page 3: “Whereas spatial encoding of time-domain information has been previously achieved for immobilized single molecules down to 4-ms time resolution based on mechanical scanning^{19,20}”

1.) The short shift time of $\sim 5\mu\text{s}$ (for 10 lines) is only advantageous if it is comparable to the illumination time (otherwise the time resolution is more or less limited by the illumination). Most researchers use illumination times of tens of milliseconds; the use of millisecond illumination is rare, sub-millisecond illumination (as used here) is hardly used. Reducing the illumination time by a factor of x also reduces the signal-to-noise ratio. This can be compensated by increasing the laser intensity by the same factor x , but only as long as the excitation intensity remains well below the saturation intensity of the fluorophore. An exposure time of $50\mu\text{s}$, as reported here, seems to be too short to be realistic. What is missing in this manuscript is: i) a rationale as to why $50\mu\text{s}$ illumination provides sufficient signal to detect single dye molecules; ii) a clear statement that the advantage of this method lies in such exceptionally short illumination times.

Response: Thank you for this discussion. The need for sub-millisecond time resolution is dependent on the application. For molecules diffusing in solution, which constitute a majority of our data, time resolutions faster than can be achieved with traditional wide-field detection are necessary. As noted, conventional wide-field single-molecule tracking has been limited to slowly diffusing bound targets. We also point out that although wide-field single-molecule measurements are often carried out at >10 ms time resolutions, which we aim to overcome in this work, in applications using a point-source detector, such as anti-Brownian electrokinetic trapping (Ref. 13) and some single-molecule FRET studies, binning detected photons into $<ms$ bins is already common.

In this revision, we quantify the SpeedyTrack photon counts and localization precisions (**Supplementary Fig. 3**). Even for exposure times as short as $50 \mu s$, we obtain localization errors ~ 30 nm, confirming our ability to perform single molecule tracking with fast time resolution. We further show with power-dependent measurements that we have not yet reached saturation. We have also performed further experiments to demonstrate the ability to obtain single-molecule FRET data at $50 \mu s$ time resolution. See our added discussion and supplementary figures on pages 6 and 7.

Page 6: “SpeedyTrack readily accesses $50 \mu s$ resolution and uniquely enables the wide-field recording of single-molecule trajectories under free, fast diffusion.”

Page 6: “While the increased temporal resolution of SpeedyTrack limits the detected photons per localization, for Cy3B we were able to collect >100 photons in $50 \mu s$ to achieve a localization precision of ~ 30 nm (**Supplementary Fig. 3**). Longer exposure times and/or higher excitation powers yielded higher photon counts and precision, and the measured localization errors at different photon counts and background levels agreed with theoretical predications³¹ after accounting for the EM-CCD multiplicative noise (**Supplementary Fig. 3**). Use of brighter fluorophores (e.g., nanoparticles) should allow higher spatial resolution.”

Page 7: “Repeating the (smFRET) measurement with a $50 \mu s$ exposure time yielded similar features (**Supplementary Fig. 6**). SpeedyTrack-smFRET thus resolved different single-molecule states at high temporal resolution.”

2.) The fluorophore density limitation should be discussed quantitatively. The method requires time traces to remain clean for the duration of the experiment, without overlap with other molecules.

Response: The reviewer is correct that the fluorophore density must be controlled to avoid overlapping trajectories. We have added related discussion to the text:

Page 4: “The typical single-molecule image density in the acquired data was ~ 100 per frame, and an upper limit of ~ 500 single-molecule images per frame is recommended to avoid overlapping of trajectories.”

Page 6: “As a limitation, SpeedyTrack requires sparse single molecules to avoid overlapping of the stretched single-molecule image trails.”

3.) In many applications in this paper, the illumination was actually not that short. For example, in Figures 2a-f and 3 the authors used $300 \mu s$. I think a shift time of $25 \mu s$ would still be OK for such a setting, which would allow standard kinetics mode readout of images of ~ 50 lines. This would still give 20 images per sequence (instead of 100 images for the SpeedyTrack), which may still be ok considering limit track length due to photobleaching. The advantage of the standard kinetics mode would be that the density of molecules can be much higher (see my point 2).

Response: We believe that the reviewer is referring to the “fast kinetics” mode, in which the camera sensor is masked to create a small effective field of view (FOV). After each exposure, charges are shifted down by the full image height h to form an image array on the chip. This approach has seen limited use in the literature, and according to our knowledge, has not been demonstrated for the SPT of freely diffusing

molecules. SpeedyTrack overcomes several major limitations of the fast kinetics mode to track freely diffusing molecules in the wide field:

Speed: Whereas the 300 μs resolution (with 25 μs shift time for a 50-line image) discussed by the reviewer may be sufficient for some applications, SpeedyTrack offers the ability to acquire data at even faster time resolutions. We have demonstrated both single-molecule tracking and single-molecule FRET at 50 μs exposure times; these time resolutions are beyond the reach of the typical fast kinetics mode.

The achievable FOV vs track length: By shifting the full FOV height h after every exposure, the FOV height competes with the track length to fill the CCD chip height. A 50-line image ($\sim 8 \mu\text{m}$) is small for imaging mammalian cells, and the limit of 20 observations translates to even shorter typical single-molecule trajectory lengths as molecules stochastically enter and leave the FOV during the experiment. Attempts to make the field larger will both further slow down acquisition and reduce track length.

By only shifting ~ 10 lines after each exposure for a wide field, SpeedyTrack provides a new scheme to not compromise between speed, FOV size, or track length. A drawback is that it requires sparse single molecules not overlapping in the stretched image trails.

We have added a brief discussion (Page 5): “Although the EM-CCD “fast kinetic mode” accesses sub-ms time resolutions, the need to shift the height of the entire field of view after each exposure seriously compromises between the achievable speed, image size, and track length, and such approaches have not been demonstrated for the SPT of freely diffusing molecules. In contrast, by shifting only ~ 10 rows per exposure, SpeedyTrack readily achieves 50 μs resolution and uniquely enables the wide-field recording of single-molecule trajectories under free, fast diffusion. ... As a limitation, SpeedyTrack requires sparse single molecules to avoid overlapping of the stretched single-molecule image trails.”

4.) A problem occurs when molecules blink, as there is an ambiguity between time and y-position. This effect is particularly problematic in single molecule localization microscopy. The authors propose a decoding scheme by introducing additional different shifts between two recorded images. I think that this decoding is only unambiguous if the shift intervals vary over the whole CCD chip; otherwise, how would one distinguish between blinks that are partially synchronized with the applied barcode (which can happen from time to time)? Moreover, decoding obviously requires additional shifts, which introduces long gaps in the image sequence, further reducing the advantage of the method over standard kinetic mode

Response: The reviewer is correct that blinking could create patterns similar to that from the variable vertical shifts. However, in our data analysis, we work with trajectories containing at least 5 observations. Using a vertical-shift pattern in which any subsequence of 4 exposures or longer is unique within the sequence, we thus built in ample fault tolerance by only keeping trajectories with Hamming distances $< 30\%$ of the length of the observed sequence (about 85% of all trajectories). We further note that in most SMLM and SPT experiments, blinking occurs on a > 10 ms timescale and so does not generally affect our ability to obtain a 5-timepoint (2.5 ms) track without blinks. The high fidelity of the reconstructed SMLM image, which resolves nanoscale ER-tubule networks, suggests negligible mismatched trajectories. The additional shifts required for this scheme indeed reduce the number of timepoints to 33 exposures per frame. Some of the above discussion was previously in the Methods section. We have improved our discussion in the text (Page 8): “The vertical-shift pattern shifted $m = 1-4$ multiples of 15 rows after each exposure to accommodate a total of 33 exposures, and was designed so that any subsequence of 4 exposures or longer is unique within the sequence (**Methods**). ... As any sequence of 4 exposures is distinct, the alignment was robust against moderate mismatches (Methods), and Dendra2 exhibits minimal single-molecule blinking at the (sub)ms time scales of our measurements³⁸.”

Response to REVIEWER COMMENTS

Reviewer #1 (No further questions.)

Reviewer #2 (Remarks to the Author):

The authors have responded satisfactorily to most of the points we raised in the first round. However, several critical issues remain inadequately addressed. These are detailed below.

1e) Accumulation of background noise

We consider that the added sentence in the main text is insufficient. The authors should discuss how the accumulation of background noise may impact the quality and interpretation of single molecule tracking data.

Response: In Supplementary Fig. 4 we show the accumulation of background noise and discuss its impact on single-molecule tracking by affecting the localization precision. The actual background level will be setup- and sample-dependent, but we have shown with our experiments that good S/N is obtained for both dyes and FPs, including the potentially more challenging case of live-cell imaging. We have improved the related discussion: “The repeated vertical shifts employed by SpeedyTrack accumulate background noise from the multiple exposures in the frame and thus reduce the achievable localization precision (Supplementary Fig. 4); the use of larger vertical shifts alleviates this issue at the expense of a decreased maximum track length (Supplementary Fig. 4).”

1g) Signal intensity measurements for individual fluorescent spots

The authors misunderstood some of the points we raised. Given that the dye-to-protein ratio determined by absorption spectroscopy was 0.9, their Cy3B-labeled carbonic anhydrase specimen is considered to include the following populations (as calculated using a Poisson distribution with a mean of 0.9): 37% of the protein molecules are tagged with one Cy3B molecule, 17% with two Cy3B molecules, and approximately 46% are unlabeled. This implies that around one-third of the observed fluorescent spots ($17/(17+37)$) represent protein tagged with two Cy3B molecules. This is a substantial fraction and raises concerns regarding the validity of referring to this as single-molecule tracking to call it single fluorescent-molecule tracking.

Since this experiment is quite straightforward, we recommend that the authors repeat the experiments using a dye-to-protein ration below 0.1 (only $\approx 5\%$ of the fluorescent spots represent 2 or more Cy3B molecules).

Note that even at a mean dye-to-protein ration of 0.4, which is a value used by the authors for Cy3B-avidin (New Supplementary Fig. 3), 17% of the fluorescent avidin spots represented 2 or 3 Cy3B molecules. This could potentially explain the broad distributions shown in Supplementary Fig. 3a.

We therefore suggest that the authors repeat the experiments shown in Supplementary Fig. 3a using a mean dye-to-protein ratio below 0.1 and compare the distribution data with the present ones obtained for the dye-to-protein ratio of 0.4. This would allow the authors to demonstrate whether it is possible to resolve the fractional contributions of monomers and dimers coexisting in the specimen, and more specifically to identify the $\approx 17\%$ fluorescent spots arising from 2 or more Cy3B molecules.

In Supplementary Fig. 3, please also indicate error bars and the number of observations.

Response: We reiterate that most of the data in our manuscript use probes that unequivocally represent a single fluorescent molecule, including free dyes, free FPs, and DNA constructs with single-dye labeling. SpeedyTrack tracked their trajectories well, even for the very fast diffusion of free Cy3B dye. It is only the Cy3B-labeled carbonic anhydrase that has a fraction being labeled by more than one dye. We further note that in single-molecule tracking, multiple dyes labeled to a single molecule does occur in some applications, and that we do not claim “single fluorescent-molecule tracking” for Cy3B-labeled carbonic anhydrase in the manuscript.

Nonetheless, to address the reviewer’s comments, we have repeated the experiments shown in Supplementary Fig. 3a using Cy3B-labeled avidin with an excessively low dye-to-protein ratio of 0.03, which we show below. We find that the photon distributions are slightly narrowed but still broader than a Poisson distribution. As discussed, single-molecule brightness is complicated by illumination patterns, and for surface-immobilized molecules, substrate interactions, including the fixation of molecular dipoles (our current setup uses linearly polarized excitation), further broaden the brightness distribution. We note that our manuscript here focuses on introducing SpeedyTrack to enable fast single-molecule tracking, and we have not tried to optimize illumination and other factors to quantify single-molecule brightness.

In Supplementary Fig. 3, each data point is calculated from 3,000-15,000 single-molecule trajectories. Errors are smaller than the symbol size. We have added this information to the caption.

(2) Diffusion data shown in Figure 2e

To demonstrate that SpeedyTrack is broadly applicable to single molecule imaging and tracking, the authors should present the photobleaching kinetics data. This could readily be obtained by using immobilized Cy3B-avidin labeled at dye-to-protein ratios less than 0.1.

Response: Thank you for this discussion. Following this suggestion, we immobilized Cy3B-avidin labeled at a 0.03 dye-to-protein ratio, and performed SpeedyTrack under our typical illumination conditions. Using an extended exposure time of 2 ms per timepoint, we recorded long single-molecule time traces to quantify the surviving fraction of emitting single molecules over time. Fitting to this result, we obtained a bleaching time constant of 110 ms (bleaching rate of 9 s^{-1}). This value is substantially longer than the duration of typical single-molecule tracks we recorded in this work, including the FRET data discussed below. We have added this result to Supplementary Fig. 2.

3b) Data analysis in Figures 3c and 3e

The analysis method based on Ref 32 (Kim et al., PNAS 2002) is, in our view, only valid under the conditions where photobleaching is negligible. We suspect that the authors’ view is that photobleaching lifetime would be negligible compared with the dwell lifetime of the molecules in the observation volume. However, under the conditions of strong excitation laser illumination, photobleaching might not be negligible. This point has to be explicitly clarified, by measuring the photobleaching lifetimes particularly for observations at higher time resolutions.

Response: Indeed our analysis assumes that photobleaching is negligible. The time-correlated conditional probability analysis we fitted (Fig. 3e) was for only 13 ms, substantially shorter than the 110-ms photobleaching time constant noted above. We have added related discussion to the manuscript.

(4) Figures 1d and 4a-c

4a) Estimation of the upper limit of the molecular densities for SMLM by VS-SpeedyTrack

As also recommended by Reviewer 3, we would feel much more comfortable if the authors could provide more quantitative estimates.

Response: We have previously provided the typical molecular densities and estimated upper limits for SpeedyTrack and VS-SpeedyTrack. The exact limits depend on the diffusivities of the species in question, the desired search radius, and the sample geometry and spatial patterns (*e.g.*, FPs inside ER lumens). The practical solution is to actively adjust the density of emitting single molecules based on the experimental data to obtain high enough counts while avoiding trajectory overlapping. This is readily achieved by titrating the sample concentration for *in vitro* samples or adjusting the power of the photoactivation laser for live-cell samples. We have added brief discussions to the text.

4b) To clarify the time courses for general readers, please also indicate the actual time in addition to the number of frames.

Response: Thank you for this suggestion. We have added the actual time to Methods and to the caption of Supplementary Fig. 9, where we split the data by frame and thus by time.

Reviewer #2 (Remarks on code availability):

The code appears to work well.

Reviewer #3 (Remarks to the Author):

The authors addressed all my points adequately. I recommend publication.

Reviewer #4 (Remarks to the Author):
